# Anytime MiniBatch: Exploiting Stragglers in Online Distributed Optimization

**Nuwan Ferdinand, Haider Al-Lawati, & Stark Draper**
Department of Electrical and Computer Engineering, University of Toronto
`{nuwan.ferdinand@,haider.al.lawati@mail.,stark.draper@}utoronto.ca`

**Matthew Nokleby**
Department of Electrical and Computer Engineering, Wayne State University, Detroit, MI
`matthew.nokleby@wayne.edu`

## ABSTRACT

Distributed optimization is vital in solving large-scale machine learning problems. A widely-shared feature of distributed optimization techniques is the requirement that all nodes complete their assigned tasks in each computational epoch before the system can proceed to the next epoch. In such settings, slow nodes, called *stragglers*, can greatly slow progress. To mitigate the impact of stragglers, we propose an online distributed optimization method called *Anytime Minibatch*. In this approach, all nodes are given a fixed time to compute the gradients of as many data samples as possible. The result is a variable per-node minibatch size. Workers then get a fixed communication time to average their minibatch gradients via several rounds of consensus, which are then used to update primal variables via dual averaging. Anytime Minibatch prevents stragglers from holding up the system without wasting the work that stragglers can complete. We present a convergence analysis and analyze the wall time performance. Our numerical results show that our approach is up to 1.5 times faster in Amazon EC2 and it is up to five times faster when there is greater variability in compute node performance.

## 1 INTRODUCTION

The advent of massive data sets has resulted in demand for solutions to optimization problems that are too large for a single processor to solve in a reasonable time. This has led to a renaissance in the study of parallel and distributed computing paradigms. Numerous recent advances in this field can be categorized into two approaches; synchronous Dekel et al. (2012); Duchi et al. (2012); Tsianos & Rabbat (2016); Zinkevich et al. (2010) and asynchronous Recht et al. (2011); Liu et al. (2015). This paper focuses on the synchronous approach. One can characterize synchronization methods in terms of the topology of the computing system, either *master-worker* or *fully distributed*. In a master-worker topology, workers update their estimates of the optimization variables locally, followed by a fusion step at the master yielding a synchronized estimate. In a fully distributed setting, nodes are sparsely connected and there is no obvious master node. Nodes synchronize their estimates via local communications. In both topologies, *synchronization* is a key step.

Maintaining synchronization in practical computing systems can, however, introduce significant delay. One cause is slow processing nodes, known as stragglers Dean et al. (2012); Yu et al. (2017); Tandon et al. (2017); Lee et al. (2018); Pan et al. (2017); S. Dutta & Nagpurkar (2018). A classical requirement in parallel computing is that all nodes process an equal amount of data per computational epoch prior to the initiation of the synchronization mechanism. In networks in which the processing speed and computational load of nodes vary greatly between nodes and over time, the straggling nodes will determine the processing time, often at a great expense to overall system efficiency. Such straggler nodes are a significant issue in cloud-based computing systems. Thus, an important challenge is the design of parallel optimization techniques that are robust to stragglers.

To meet this challenge, we propose an approach that we term *Anytime MiniBatch* (AMB). We consider a fully distributed topologyand consider the problem of stochastic convex optimization

via dual averaging Nesterov (2009); Xiao (2010). Rather than fixing the minibatch size, we fix the computation time ($T$) in each epoch, forcing each node to "turn in" its work after the specified fixed time has expired. This prevents a single straggler (or stragglers) from holding up the entire network, while allowing nodes to benefit from the partial work carried out by the slower nodes. On the other hand, fixing the computation time means that each node process a different amount of data in each epoch. Our method adapts to this variability. After computation, all workers get fixed communication time ($T_c$) to share their gradient information via averaging consensus on their dual variables, accounting for the variable number of data samples processed at each node. Thus, the epoch time of AMB is fixed to $T + T_c$ in the presence of stragglers and network delays.

We analyze the convergence of AMB, showing that the online regret achieves $\mathcal{O}(\sqrt{\bar{m}})$ performance, which is optimal for gradient based algorithms for arbitrary convex loss Dekel et al. (2012). In here, $\bar{m}$ is the expected sum number of samples processed across all nodes. We further show an upper bound that, in terms of the expected *wall time* needed to attain a specified regret, AMB is $\mathcal{O}(\sqrt{n-1})$ faster than methods that use a fixed minibatch size under the assumption that the computation time follows an arbitrary distribution where $n$ is the number of nodes. We provide numerical simulations using Amazon Elastic Compute Cloud (EC2) and show that AMB offers significant acceleration over the fixed minibatch approach.

## 2 RELATED WORK

This work contributes to the ever-growing body of literature on distributed learning and optimization, which goes back at least as far as Tsitsiklis et al. (1986), in which distributed first-order methods were considered. Recent seminal works include Nedic & Ozdaglar (2009), which considers distributed optimization in sensor and robotic networks, and Dekel et al. (2012), which considers stochastic learning and prediction in large, distributed data networks. A large body of work elaborates on these ideas, considering differences in topology, communications models, data models, etc. Duchi et al. (2012); Tsianos et al. (2012); Shi et al. (2015); Xi & Khan (2017). The two recent works most similar to ours are Tsianos & Rabbat (2016) and Nokleby & Bajwa (2017), which consider distributed online stochastic convex optimization over networks with communications constraints. However, both of these works suppose that worker nodes are homogeneous in terms of processing power, and do not account for the straggler effect examined herein. The recent work Pan et al. (2017); Tandon et al. (2017); S. Dutta & Nagpurkar (2018) proposed synchronous fixed minibatch methods to mitigate stragglers for master-worker setup. These methods either ignore stragglers or use redundancy to accelerate convergence in the presence of stragglers. However, our approach in comparison to Pan et al. (2017); Tandon et al. (2017); S. Dutta & Nagpurkar (2018) utilizes work completed by both fast and slow working nodes, thus results in faster wall time in convergence.

## 3 SYSTEM MODEL AND ALGORITHM

In this section we outline our computation and optimization model and step through the three phases of the AMB algorithm. The pseudo code of the algorithm is provided in App. A. We defer discussion of detailed mathematical assumptions and analytical results to Sec. 4.

We suppose a computing system that consists of $n$ compute nodes. Each node corresponds to a vertex in a connected and undirected graph $G(V, E)$ that represents the inter-node communication structure. The vertex set $V$ satisfies $|V| = n$ and the edge set $E$ tells us which nodes can communicate directly. Let $\mathcal{N}_i = \{j \in V : (i, j) \in E, i \neq j\}$ denote the neighborhood of node $i$.

The collaborative objective of the nodes is to find the parameter vector $w \in W \subseteq \mathbb{R}^d$ that solves

$$w^* = \arg \min_{w \in W} F(w) \quad \text{where} \quad F(w) := \mathbb{E}_x[f(w, x)]. \tag{1}$$

The expectation $\mathbb{E}_x[\cdot]$ is computed with respect to an unknown probability distribution $Q$ over a set $X \subseteq \mathbb{R}^d$. Because the distribution is unknown, the nodes must approximate the solution in (1) using data points drawn in an independent and identically distributed (i.i.d.) manner from $Q$.

AMB uses dual averaging Nesterov (2009); Dekel et al. (2012) as its optimization workhorse and averaging consensus Nokleby & Bajwa (2017); Tsianos & Rabbat (2016) to facilitate collaboration among nodes. It proceeds in epochs consisting of three phases: *compute*, in which nodes compute

local minibatches; *consensus*, in which nodes average their dual variables together; and *update*, in which nodes take a dual averaging step with respect to the consensus-averaged dual variables. We let $t$ index each epoch, and each node $i$ has a primal variable $w_i(t) \in \mathbb{R}^d$ and dual variable $z_i(t) \in \mathbb{R}^d$. At the start of the first epoch, $t = 1$, we initialize all primal variables to the same value $w(1)$ as

$$w_i(1) = w(1) = \arg\min_{w \in W} h(w), \tag{2}$$

and all dual variables to zero, i.e., $z_i(1) = 0 \in \mathbb{R}^d$. In here, $h : W \to \mathbb{R}$ is a 1-strongly convex function.

**Compute Phase:** All workers are given $T$ *fixed time* to compute their local minibatches. During each epoch, each node is able to compute $b_i(t)$ gradients of $f(w, x)$, evaluated at $w_i(t)$ where the data samples $x_i(t, s)$ are drawn i.i.d. from $Q$. At the end of epoch $t$, each node $i$ computes its local minibatch gradient:

$$g_i(t) = \frac{1}{b_i(t)} \sum_{s=1}^{b_i(t)} \nabla_w f\big(w_i(t), x_i(t, s)\big). \tag{3}$$

As we fix the compute time, the local minibatch size $b_i(t)$ is a random variable. Let $b(t) := \sum_{i=1}^{n} b_i(t)$ be the *global* minibatch size aggregated over all nodes. This contrasts with traditional approaches in which the minibatch is fixed. In Sec. 4 we provide a convergence analysis that accounts for the variability in the amount of work completed by each node. In Sec. 5, we presents a wall time analysis based on random local minibatch sizes.

**Consensus Phase:** Between computational epochs each node is given a fixed amount of time, $T_c$, to communicate with neighboring nodes. The objective of this phase is for each node to get (an approximation of) the following quantity:

$$\frac{1}{b(t)} \sum_{i=1}^{n} b_i(t)[z_i(t) + g_i(t)] = \frac{1}{b(t)} \sum_{i=1}^{n} b_i(t)z_i(t) + \frac{1}{b(t)} \sum_{i=1}^{n} \sum_{s=1}^{b_i(t)} \nabla_w f\big(w_i(t), x_i(t, s)\big)$$

$$:= \bar{z}(t) + g(t). \tag{4}$$

The first term, $\bar{z}(t)$, is the weighted average of the previous dual variables. The second, $g(t)$, is the average of all gradients computed in epoch $t$.

The nodes compute this quantity approximately via several synchronous rounds of average consensus. Each node waits until it hears from all neighbors before starting a consensus round. As we have fixed communication time $T_c$, the number of consensus rounds $r_i(t)$ varies across workers and epochs due to random network delays. Let $P$ be a positive semi-definite, doubly-stochastic matrix (i.e., all entries of $P$ are non-negative and all row- and column-sums are one) that is consistent with the graph $G$ (i.e., $P_{i,j} > 0$ only if $i = j$ or if $(i, j) \in E$). At the start of the consensus phase, each node $i$ shares its message $m_i^{(0)} = nb_i(t)[z_i(t) + g_i(t)]$ with its neighboring nodes. Let $[r]$ stand for $[r] \in \{1, \ldots r\}$. Then, in consensus iteration $k \in [r_i(t)]$ node $i$ computes its update as

$$m_i^{(k)} = \sum_{j=1}^{n} P_{i,j} m_j^{(k-1)} = \sum_{j=1}^{n} (P_{i,j})^k m_j^{(0)}.$$

As long as $G$ is connected and the second-largest eigenvalue of $P$ is strictly less than unity, the iterations are guaranteed to converge to the true average. For finite $r_i(t)$, each node will have an error in its approximation. Instead of (4), at the end of the rounds of consensus, node $i$ will have

$$z_i(t + 1) = \bar{z}(t) + g(t) + \xi_i(t), \tag{5}$$

where $\xi_i(t)$ is the error. We use $\mathcal{D}^{(r_i(t))}\big(\{y_j\}_{j \in V}, i\big)$ to denote the distributed averaging affected by $r_i(t)$ rounds of consensus. Thus,

$$z_i(t + 1) = \frac{1}{b(t)} \mathcal{D}^{(r_i(t))}\Big(\big\{nb_j(t)\big[z_j(t) + g_j(t)\big]\big\}_{j \in V}, i\Big) = \frac{1}{b(t)} m_i^{(r_i(t))}. \tag{6}$$

We note that the updated dual variable $z_i(t + 1)$ is a normalized version of the distributed average solution, normalized by $b(t) = \sum_{i=1}^{n} b_i(t)$.

**Update Phase:** After distributed averaging of dual variables, each node updates its primal variable as

$$w_i(t+1) = \arg\min_{w \in W} \Big\{ \langle w, z_i(t+1) \rangle + \beta(t+1)h(w) \Big\}; \tag{7}$$

where $\langle \cdot, \cdot \rangle$ denotes the standard inner product. As will be discussed further in our analysis, in this paper we assume $h : W \rightarrow \mathbb{R}$ to be a 1-strongly convex function and $\beta(t)$ to be a sequence of positive non-decreasing parameters, i.e., $\beta(t) \leq \beta(t+1)$. We also work in Euclidean space where $h(w) = \|w\|^2$ is a typical choice.

## 4 ANALYSIS

In this section we analyze the performance of AMB in terms of expected regret. As the performance is sensitive to the specific distribution of the processing times of the computing platform used, we first present a generic analysis in terms of the number of epochs processed and the size of the minibatches processed by each node in each epoch. Then in Sec. 5, in order to illustrate the advantages of AMB, we assert a probabilistic model on the processing time and analyze the performance in terms of the elapsed "wall time" .

### 4.1 PRELIMINARIES

We assume that the feasible space $W \in \mathbb{R}^d$ of the primal optimization variable $w$ is a closed and bounded convex set where $D = \max_{w,u \in W} \|w - u\|$. Let $\| \cdot \|$ denote the $\ell_2$ norm. We assume the objective function $f(w, x)$ is convex and differentiable in $w \in W$ for all $x \in X$. We further assume that $f(w, x)$ is Lipschitz continuous with constant $L$, i.e.

$$|f(w, x) - f(\tilde{w}, x)| \leq L\|w - \tilde{w}\|, \quad \forall x \in X, \text{and } \forall w, \tilde{w} \in W. \tag{8}$$

Let $\nabla f(w, x)$ be the gradient of $f(w, x)$ with respect to $w$. We assume the gradient of $f(w, x)$ is Lipschitz continuous with constant $K$, i.e.,

$$\|\nabla f(w, x) - \nabla f(\tilde{w}, x)\| \leq K\|w - \tilde{w}\|, \quad \forall x \in X, \text{and } \forall w, \tilde{w} \in W. \tag{9}$$

As mentioned in Sec. 3,

$$F(w) = \mathbb{E}[f(w, x)], \tag{10}$$

where the expectation is taken with respect to the (unknown) data distribution $Q$, and thus $\nabla F(w) = \mathbb{E}[\nabla f(w, x)]$. We also assume that there exists a constant $\sigma$ that bounds the second moment of the norm of the gradient so that

$$\mathbb{E}[\|\nabla f(w, x) - \nabla F(w)\|^2] \leq \sigma^2, \forall x \in X, \text{and } \forall w \in W. \tag{11}$$

Let the global minimum be denoted $w^* := \arg\min_{w \in W} F(w)$.

### 4.2 SAMPLE PATH ANALYSIS

First we bound the consensus errors. Let $z(t)$ be the exact dual variable without any consensus errors at each node

$$z(t) = \bar{z}(t-1) + g(t-1). \tag{12}$$

The following Lemma bounds the consensus errors, which is obtained using (Tsianos & Rabbat, 2016, Theorem 2)

**Lemma 1** *Let $z_i^{(r)}(t)$ be the output after $r$ rounds consensus. Let $\lambda_2(P)$ be the second eigenvalue of the matrix $P$ and let $\epsilon \geq 0$, then*

$$\|z_i^{(r)}(t) - z(t)\| \leq \epsilon, \quad \forall i \in [n], t \in [\tau], \tag{13}$$

*if the number of consensus rounds satisfies*

$$r_i(t) \geq \left\lceil \frac{\log\left(2\sqrt{n}(1 + 2L/\epsilon)\right)}{1 - \lambda_2(P)} \right\rceil, \quad \forall i \in [n], t \in [\tau]. \tag{14}$$

We characterize the regret after $\tau$ epochs, averaging over the data distribution but keeping a fixed "sample path" of per-node minibatch sizes $b_i(t)$. We observe that due to the time spent in communicating with other nodes via consensus, each node has computation cycles that could have been used to compute more gradients had the consensus phase been shorter (or nonexistent). To model this, let $a_i(t)$ denote the number of *additional* gradients that node $i$ *could have* computed had there been no consensus phase. This undone work does not impact the system performance, but does enter into our characterization of the regret. Let $c_i(t) = b_i(t) + a_i(t)$ be the total number of gradients that node $i$ had the potential to compute during the $t$-th epoch. Therefore, the total potential data samples processed in the $t$-th epoch is $c(t) = \sum_{i=1}^{n} c_i(t)$. After $\tau$ epochs the total number of data points that could have been processed by all nodes in the absence of communication delays is

$$m = \sum_{t=1}^{\tau} c(t). \tag{15}$$

An important quantity is the ratio of total *potential* computations in each epoch to that actually completed. Define the maximum such minibatch "skewness" as

$$\gamma = \max_{t \in [\tau-1]} \frac{c(t+1)}{b(t)}. \tag{16}$$

It turns out that it is important to compute this skewness across epochs (i.e., $c(t+1)$ versus $b(t)$) in order to bound the regret via a telescoping sum. [Details can be found in the supplementary material.]

In practice, $a_i(t)$ and $b_i(t)$ (and therefore $c_i(t)$) depend on latent effects, e.g., how many other virtual machines are co-hosted on node $i$, and therefore we model them as random variables. We bound the expected regret for a fixed sample path of $a_i(t)$ and $b_i(t)$. The sample paths of importance are $c_{\text{tot}}(\tau) = \{c_i(t)\}_{i \in V, t \in [\tau]}$ and $b_{\text{tot}}(\tau) = \{b_i(t)\}_{i \in V, t \in [\tau]}$, where we introduce $c_{\text{tot}}$ and $b_{\text{tot}}$ for notational compactness.

Define the average regret after $\tau$ epochs as

$$R(\tau) = \mathbb{E}\big[R|b_{\text{tot}}(\tau), c_{\text{tot}}(\tau)\big] = \mathbb{E}\left[\sum_{t=1}^{\tau}\sum_{i=1}^{n}\sum_{s=1}^{c_i(t)}\Big[f\big(w_i(t), x_i(t,s)\big) - F(w^*)\Big]\right], \tag{17}$$

where the expectation is taken with respect the the i.i.d. sampling from the distribution $Q$. Then, we have the following bound on $R(\tau)$.

**Theorem 2** *Suppose workers collectively processed $m$ samples after $\tau$ epochs, cf. (15), minibatch skewness parameter $\gamma$, cf. (16), and let $c_{\max} = \max_{t \in [\tau]} c(t)$, $c_{\text{avg}} = (1/\tau)\sum_{t=1}^{\tau} c(t)$ and $\delta = \max_{\{t,t'\} \in \{1,\tau-1\}} |c(t) - c(t')|$ be the maximum, average, and variation across $c(t)$. Further, suppose the averaging consensus has additive accuracy $\epsilon$, cf. Lemma 1. Then, the expected regret is*

$$R(\tau) \leq c_{\max}[F(w(1)) - F(w^*) + \beta(\tau)h(w^*)] + \frac{3K^2\epsilon^2 c_{\max}\mu^{3/2}}{4}$$
$$+ \delta\left(\left(1 + \frac{\tau}{2}\right)F(w^*) + h(w^*)\left(K + \tau^{1/2}c_{\text{avg}}^{-1/2}\right)\right)\tau$$
$$+ \left(2KD\epsilon + \frac{\gamma\sigma^2}{2} + 2L\epsilon c_{\max} + 2\delta KD\epsilon\tau\right)\sqrt{m}. \tag{18}$$

Theorem 2 is proved in App. B of the supplementary material.

We now make a few comments about this result. First, recall that the expectation is taken with respect to the data distribution, but holds for any sample path of minibatch sizes. Further, the regret bound depends only on the summary statistics $c_{\max}$, $c_{\text{avg}}$, $\delta$, and $\gamma$. These parameters capture the distribution of the processing speed at each node. Further, the impact of consensus error, which depends on the communication speed relative to the processing speed of each node, is summarized in the assumption of uniform accuracy $\epsilon$ on the distributed averaging mechanism. Thus, Theorem 2 is a sample path result that depends only coarsely on the distribution of the speed of data processing.

Next, observe that the dominant term is the final one, which scales in the aggregate number of samples $m$. The first term is approximately constant, only scaling with the monotonically increasing $\beta$ and

$c_{\max}$ parameters. The terms containing $\epsilon$ characterizes the effect of imperfect consensus, which can be reduced by increasing the number of rounds of consensus. The effect of variability across $c(t)$ is reflected in the terms containing the $c_{\max}$, $c_{\text{avg}}$ and $\delta$ parameters. If perfect consensus were achieved ($\epsilon = 0$) then all components of the final term that scales in $\sqrt{m}$ would disappear except for the term that contains the minibatch skewness parameter $\gamma$. It is through this term that the amount of useful computation performed in each epoch ($b_i(t) \leq c_i(t)$) enters the result.

In the special case of constant minibatch size $c_{\max} = c_{\text{avg}}$ and $\delta = 0$, we have the following corollary.

**Corollary 3** *If $c(t) = c$ for all $t \in [\tau]$ and the consensus error $\epsilon \leq 1/c$, then the expected regret is*

$$R(\tau) = \mathcal{O}(c + \sqrt{m}). \tag{19}$$

*Furthermore, if $c = m^\rho$ for a constant $\rho \in (0, 1/2]$, then $R(\tau) = \mathcal{O}(\sqrt{m})$.*

### 4.3 EXPECTED REGRET ANALYSIS

We can translate Theorem 2 and Cor. 3 to a regret bound averaged over the sample path. Since the summary statistics $c_{\max}$, $c_{\text{avg}}$, $\delta$, and $\gamma$ are sufficient to bound the regret, we assert a joint distribution $p$ over these terms rather than over the sample path $b_{\text{tot}}(\tau), c_{\text{tot}}(\tau)$. For the following result, we need only specify several moments of the distribution. In Sec. 5 we will take the further step of choosing a specific distribution $p$.

**Theorem 4** *Let $\bar{c} = \mathbb{E}_p[c(t)]$ so that $\bar{m} = \tau \bar{c}$ is the expected total work that can be completed in $\tau$ epochs. Also, let $1/\hat{b} = \mathbb{E}_p[1/b(t)]$. If averaging consensus has additive accuracy $\epsilon$, then the expected regret is bounded by*

$$\mathbb{E}_p[R(\tau)] \leq \bar{c}\big[F(w(1)) - F(w^*) + \bar{\beta}(\tau)h(w^*)\big] + \frac{3K^2\epsilon^2\bar{c}^{5/2}}{4} + \left(2KD\epsilon + \frac{\bar{c}\sigma^2}{2\hat{b}} + 2L\epsilon\bar{c}\right)\sqrt{\bar{m}}.$$

Theorem 4 is proved in App. F of the supplementary material. Note that this expected regret is over both the i.i.d. choice of data samples and the i.i.d. choice of $(b(t), c(t))$ pairs.

**Corollary 5** *If $\epsilon \leq 1/\bar{c}$, the expected regret is*

$$\mathbb{E}_p[R(\tau)] \leq \mathcal{O}(\bar{c} + \sqrt{\bar{m}}). \tag{20}$$

*Further, if $\bar{c} = \bar{m}^\rho$ for a constant $\rho \in (0, 1/2)$, then $\mathbb{E}[R] \leq \mathcal{O}(\sqrt{\bar{m}})$.*

**Remark 1** *Note that by letting $\epsilon = 0$, we can immediately find the results for master-worker setup.*

## 5 WALL TIME ANALYSIS

In the preceding section we studied regret as a function of the number of epochs. The advantages of AMB is the reduction of wall time. That is, AMB can get to same convergence in less time than fixed minibatch approaches. Thus, in this section, we caracterize the wall time performance of AMB.

In AMB, each epoch corresponds to a fixed compute time $T$. As we have already commented, this contrasts with fixed minibatch approaches where they have variable computing times. We refer "Fixed MiniBatch" methods as FMB. To gain insight into the advantages of AMB, we develop an understanding of the regret per unit time.

We consider an FMB method in which each node computes computes $b/n$ gradients, where $b$ is the size of the global minibatch in each epoch. Let $T_i(t)$ denote the amount of time taken by node $i$ to compute $b/n$ gradients for FMB method. We make the following assumptions:

**Assumption 1** *The time $T_i(t)$ follows an arbitrary distribution with the mean $\mu$ and the variance $\sigma^2$. Further, $T_i(t)$ is identical across node index $i$ and epoch index $t$. .*

**Assumption 2** *If node $i$ takes $T_i(t)$ seconds to compute $b/n$ gradients in the $t$-th epoch, then it will take $nT_i(t)/b$ seconds to compute one gradient.*

**Lemma 6** *Let Assumptions 1 and 2 hold. Let the FMB scheme have a minibatch size of $b$. Let $\bar{b}$ be the expected minibatch size of AMB. Then, if we fix the computation time of an epoch in AMB to $T = (1 + n/b)\mu$, we have $\bar{b} \geq b$.*

Lemma 6 is proved in App. G and it shows that the expected minibatch size of AMB is at least as big as FMB if we fix $T = (1 + n/b)\mu$. Thus, we get same (or better) expected regret bound. Next, we show that AMB achieve this in less time.

**Theorem 7** *Let Assumptions 1 and 2 hold. Let $T = (1 + n/b)\mu$ and minibatch size of FMB is $b$. Let $S_A$ and $S_F$ be the total compute time across $\tau$ epochs of AMB and FMB, respectively, then*

$$S_F \leq \left(1 + \frac{\sigma}{\mu}\sqrt{n-1}\right) S_A. \tag{21}$$

The proof is given in App. G. Lemma 6 and Theorem 7 show that our method attains the same (or better) bound on the expected regret that is given in Theorem 4 but is at most $\left(1 + \sigma/\mu\sqrt{n-1}\right)$ faster than traditional FMB methods. In Bertsimas et al. (2006), it was shown this bound is tight and there is a distribution that achieves it. In our setup, there are no analytical distributions that exactly match with finishing time distribution. Recent papers on stragglers Lee et al. (2018); S. Dutta & Nagpurkar (2018) use the shifted exponential distribution to model $T_i(t)$. The choice of shifted exponential distribution is motivated by the fact that it strikes a good balance between analytical tractability and practical behavior. Based on the assumption of shifted exponential distribution, we show that AMB is $\mathcal{O}(\log(n))$ faster than FMB. This result is proved in App. H.

## 6 NUMERICAL EVALUATION

To evaluate the performance of AMB and compare it with that of FMB, we ran several experiments on Amazon EC2 for both schemes to solve two different classes of machine learning tasks: linear regression and logistic regression using both synthetic and real datasets. In this section we present error vs. wall time performance using two experiments. Additional simulations are given in App. I

### 6.1 DATASETS

We solved two problems using two datasets: synthetic and real. Linear regression problem was solved using synthetic data. The element of global minimum parameter, $w^* \in \mathbb{R}^d$, is generated from the multivariate normal distribution $\mathcal{N}(\mathbf{0}, \mathbf{I})$. The workers observe a sequence of pairs $(x_i(s), y_i(s))$ where $s$ is the time index, data $x_i(s) \in \mathbb{R}^d$ are i.i.d. $\mathcal{N}(\mathbf{0}, \mathbf{I})$, and the labels $y_i(s) \in \mathbb{R}$ such that $y_i(s) = x_i(s)^T w^* + \eta_i(s)$. The additive noise sequence $\eta_i(s) \in \mathbb{R}$ is assumed to be i.i.d $\mathcal{N}(0, 10^{-3})$. The aim of all nodes is to collaboratively learn the true parameter $w^*$. The data dimension is $d = 10^5$.

For the logistic regression problem, we used the MNIST images of numbers from 0 to 9. Each image is of size $28 \times 28$ pixels which can be represented as a 784-dimensional vector. We used MNIST training dataset that consists of 60,000 data points. The cost function is the cross-entropy function $J$

$$J(y) = -\sum_i \mathbb{1}[y = i]\mathbb{P}(y = i|x) \tag{22}$$

where $x$ is the observed data point sampled randomly from the dataset, $y$ is the true label of $x$. $\mathbb{1}[.]$ is the indicator function and $\mathbb{P}(y = i|x)$ is the predicted probability that $y = i$ given the observed data point $x$ which can be calculated using the softmax function. In other words, $\mathbb{P}(y = i|x) = e^{w_i x}/\sum_j e^{w_j x}$. The aim of the system is to collaboratively learn the parameter $w \in \mathbb{R}^{c \times d}$, where $c = 10$ classes and $d = 785$ the dimension (including the bias term) that minimizes the cost function while streaming the inputs $x$ online.

### 6.2 EXPERIMENTS ON EC2

We tested the performance of AMB and FMB schemes using fully distributed setup. We used a network consisting of $n = 10$ nodes, in which the underlying network topology is given in Figure 2 of App. I.1. In all our experiments, we used t2.micro instances and ami-6b211202, a publicly available

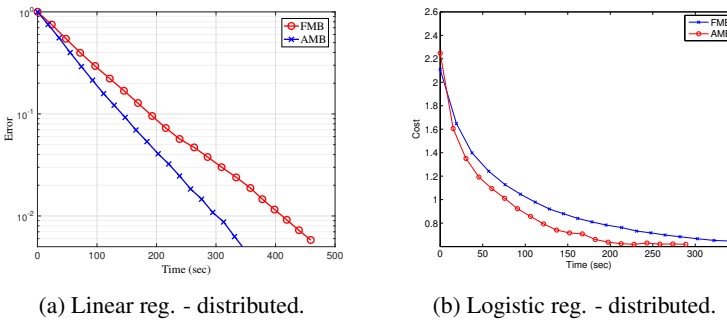

(a) Linear reg. - distributed.          (b) Logistic reg. - distributed.

Figure 1: AMB vs. FMB performance comparison on EC2.

Amazon Machine Image (AMI), to launch the instances. Communication between nodes were handled through Message Passing Interface (MPI).

To ensure a fair comparison between the two schemes, we ran both algorithms repeatedly and for a long time and averaged the performance over the same duration. We also observed that the processors finish tasks much faster during the first hour or two before slowing significantly. After that initial period, workers enter a steady state in which they keep their processor speed relatively constant except for occasional bursts. We discarded the transient behaviour and considered the performance during the steady-state.

### 6.2.1 LINEAR REGRESSION

We ran both AMB and FMB in a fully distributed setting to solve the linear regression problem. In FMB, each worker computed $b = 6000$ gradients. The average compute time during the steady-state phase was found to be $14.5$ sec. Therefore, in AMB case, the compute time for each worker was set to be $T = 14.5$ sec. and we set $T_c = 4.5$ sec. Workers are allowed $r = 5$ average rounds of consensus to average their calculated gradients.

Figure 1(a) plots the error vs. wall time, which includes both computation and communication times. One can notice AMB clearly outperforms FMB. In fact, the total amount of time spent by FMB to finish all the epochs is larger than that spent by AMB by almost 25% as shown in Figure 1(a) (e.g., the error rate achieved by FMB after 400 sec. has already been achieved by AMB after around 300 sec.). We notice, both scheme has the same average inter-node communication times. Therefore, when ignoring inter-node communication times, this ratio increases to almost 30%.

### 6.2.2 LOGISTIC REGRESSION

In here we perform logistic regression using $n = 10$ distributed nodes. The network topology is as same as above. The per-node fixed minibatch in FMB is $b/n = 800$ while the fixed compute time in AMB is $T = 12$ sec. and the communication time $T_c = 3$ sec. As in the linear regression experiment above, the workers on average go through $r = 5$ round of consensus.

Figures 1(b) shows the achieved cost vs. wall clock time. We observe AMB outperforms FMB by achieving the same error rate earlier. In fact, Figure 1(b) demonstrates that AMB is about 1.7 times faster than FMB. For instance, the cost achieved by AMB at 150 sec. is almost the same as that achieved by FMB at around 250 sec.

## 7 CONCLUSION

We proposed a distributed optimization method called Anytime MiniBatch. A key property of our scheme is that we fix the computation time of each distributed node instead of minibatch size. Therefore, the finishing time of all nodes are deterministic and does not depend on the slowest processing node. We proved the convergence rate of our scheme in terms of the expected regret bound. We performed numerical experiments using Amazon EC2 and showed our scheme offers significant improvements over fixed minibatch schemes.

# 8 ACKNOWLEDGMENT

This work was supported by the National Science Foundation (NSF) under Grant CCF-1217058, by the Natural Science and Engineering Research Council (NSERC) of Canada, including through a Discover Research Grant, by the NSERC Postdoctoral Fellowship, and by the Oman Government Post Graduate Scholarship

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

## A  AMB Algorithm

The pseudocode of the Anytime Minibatch scheme operating in a distributed setting is given in Algorithm 1. Line 2 is for initialization purpose. Lines $3-8$ corresponds to the compute phase during which each node $i$ calculates $b_i(t)$ gradients. The consensus phase steps are given in lines $9-21$. Each node first averages the gradients (line 9) and calculates the initial messages $m_i(t)$ it will share with its neighbours (line 10). Lines $14-19$ corresponds to the communication rounds that results in distributed averaging of the dual variable $z_i(t+1)$ (line 21). Finally, line 22 represents the update phase in which each node updates its primal variable $w_i(t+1)$.

For the hub-and-spoke configuration, one can easily modify the algorithm as only a single consensus round is required during which all workers send their gradients to the master node which calculates $z(t+1)$ and $w(t+1)$ followed by a communication from the master to the workers with the updated $w(t+1)$.

---

**Algorithm 1** AMB Algorithm

---

1: **for all** $t = 1, 2, ...$ **do**
2:      initialize $g_i(t) = 0, b_i(t) = 0$
3:      $T_0 = current\_time$
4:      **while** $current\_time - T_0 \leq T$ **do**
5:          receive input $x_i(t, s)$ sampled i.i.d. from $Q$
6:          calculate $g_i(t) = g_i(t) + \nabla f(w_i(t), x_i(t, s))$
7:          $b_i(t) + +$
8:      **end while**
9:      start consensus rounds
10:      $g_i(t) = \frac{1}{b_i(t)} g_i(t)$
11:      $m_i^{(0)} = n b_i(t)[z_i(t) + g_i(t)]$
12:      set $k = 0$
13:      $T_1 = current\_time$
14:      **while** $current\_time - T_1 \leq T_c$ **do**
15:          receive $m_j^k, \forall j \in \mathcal{N}_i$
16:          $m_i^{(k+1)} = \sum_{j \in \mathcal{N}_i} P_{i,j} m_j^k$
17:          send $m_i^{(k+1)}$ to all nodes in $\mathcal{N}_i$
18:          $k + +$
19:      **end while**
20:      $r_i(t) = k$
21:      $z_i(t + 1) = \frac{1}{b(t)} m_i^{(r_i(t))}$
22:      $w_i(t + 1) = \arg\min_{w \in W} \left\{ \langle w, z_i(t + 1) \rangle + \beta(t + 1) h(w) \right\}$
23: **end for**

---

## B  Proof of Theorem 2

In this section, we prove Theorem 2. There are three factors impacting the convergence of our scheme; first is that gradient is calculated with respect to $f(w, x)$ rather than directly computing the exact gradient $\nabla_w F(w)$, the second factor is the errors due to limited consensus rounds, and the last factor is that we have variable sized minibatch size over epochs. We bound these errors to find the expected regret bound with respect to a sample path.

Let $w(t)$ be the primal variable computed using the exact dual $z(t)$, cf. 12:

$$w(t) = \arg\min_{w \in W} \{ \langle w, z(t) \rangle + \beta(t) h(w) \} \tag{23}$$

From (Tsianos & Rabbat, 2016, Lemma 2), we have

$$\|w_i(t) - w(t)\| \leq \frac{1}{\beta(t)}\|z_i(t) - z(t)\|, \quad \forall i \in [n]$$

$$\leq \frac{\epsilon}{\beta(t)}. \tag{24}$$

Recall that $z_i(t)$ is the dual variable after $r$ rounds of consensus. The last step is due to Lemma 1. Let $X(t)$ be the total set of samples processed by the end of $t$-th epoch:

$$X(t) := \{x_i(t', s) : i \in [n], t' \in [t], s \in [c(t)]\}.$$

Let $\mathbb{E}[\cdot]$ denote the expectation over the data set $X(\tau)$ where we recall $\tau$ is the number of epochs. Note that conditioned on $X(t-1)$ the $w_i(t)$ and $x_i(t, s)$ are independent according to equation 7. Thus,

$$\mathbb{E}\left[f\left(w_i(t), x_i(t, s)\right)\right] = \mathbb{E}\left[\mathbb{E}_{x_i(t,s)}\left[f\left(w_i(t), x_i(t, s)\right) | X(t-1)\right]\right]$$

$$= \mathbb{E}\left[F(w_i(t))\right]. \tag{25}$$

where equation 25 is due to equation 10. From equation 17 we have

$$\mathbb{E}\left[R|b_{\text{tot}}(\tau), c_{\text{tot}}(\tau)\right] = \sum_{t=1}^{\tau}\sum_{i=1}^{n}\sum_{s=1}^{c_i(t)}\mathbb{E}\left[f\left(w_i(t), x_i(t, s)\right) - F(w^*)\right] \tag{26}$$

$$= \sum_{t=1}^{\tau}\sum_{i=1}^{n}\sum_{s=1}^{c_i(t)}\mathbb{E}\left[F(w_i(t)) - F(w^*)\right]. \tag{27}$$

Now, we add and subtract $F(w(t))$ from equation 17 to get

$$\mathbb{E}[R|b_{\text{tot}}(\tau), c_{\text{tot}}(\tau)] = \sum_{t=1}^{\tau}\sum_{i=1}^{n}\sum_{s=1}^{c_i(t)}\mathbb{E}\left[F(w(t)) - F(w^*) + F(w_i(t)) - F(w(t))\right]$$

$$= \sum_{t=1}^{\tau}c(t)\mathbb{E}[F(w(t)) - F(w^*)] + \sum_{t=1}^{\tau}\sum_{i=1}^{n}c_i(t)\mathbb{E}[F(w_i(t)) - F(w(t))]$$

$$\leq \sum_{t=1}^{\tau}c(t)\mathbb{E}[F(w(t)) - F(w^*)] + \sum_{t=1}^{\tau}\sum_{i=1}^{n}c_i(t)L\mathbb{E}[\|w_i(t) - w(t)\|] \tag{28}$$

$$\leq \sum_{t=1}^{\tau}c(t)\mathbb{E}[F(w(t)) - F(w^*)] + \sum_{t=1}^{\tau}\sum_{i=1}^{n}\frac{c_i(t)L\epsilon}{\beta(t)} \tag{29}$$

$$= \sum_{t=1}^{\tau}c(t)\mathbb{E}[F(w(t)) - F(w^*)] + L\epsilon\sum_{t=1}^{\tau}\frac{c(t)}{\beta(t)}. \tag{30}$$

Note that equation 28 and equation 29 are due to equation 8 and equation 24. Now, we bound the first term in the following Lemma, which is proved in App. C.

**Lemma 8** *Let* $\beta(t) = K + \alpha(t)$ *where* $\alpha(t) = \sqrt{\frac{t}{\mu}}$. *Then*

$$\sum_{t=1}^{\tau}c(t)\mathbb{E}\left[F(w(t)) - F(w^*)\right] \leq c(1)[F(w(1)) - F(w^*)] + \sum_{t=2}^{\tau}(c(\tau) - c(t))F(w^*)$$

$$+ c(\tau)\beta(\tau)h(w^*) + \sum_{t=1}^{\tau-1}\frac{KD\epsilon}{\beta(t)} + \sum_{t=1}^{\tau-1}\frac{c(t+1)\sigma^2}{4b(t)\alpha(t)} + \frac{K^2\epsilon^2}{4}\sum_{t=1}^{\tau-1}\frac{c(t+1)}{\alpha(t)\beta(t)^2} + \mathbb{E}[\psi] \tag{31}$$

*where*

$$\mathbb{E}[\psi] \leq \sum_{t=1}^{\tau-1}(c(t) - c(t+1))\left((t-1)F(w^*) + \beta(t)h(w^*) + 2KD\epsilon\sqrt{\mu t}\right). \tag{32}$$

In equation 31, the first term is a constant, which depends on the initialization. The fourth and the sixth terms are due to consensus errors and the fifth term is due to noisy gradient calculation. The second and the last term $\mathbb{E}[\psi]$ are due to variable minibatch sizes.

Now, the total regret can be obtained by using Lemma 8 in equation 30

$$\mathbb{E}[R|b_{\text{tot}}(\tau), c_{\text{tot}}(\tau)] \leq c(1)[F(w(1)) - F(w^*)] + \sum_{t=2}^{\tau}(c(\tau) - c(t))F(w^*) + c(\tau)\beta(\tau)h(w^*)$$

$$+ \sum_{t=1}^{\tau-1}\frac{KD\epsilon}{\beta(t)} + + \sum_{t=1}^{\tau-1}\frac{c(t+1)\sigma^2}{4b(t)\alpha(t)} + \frac{K^2\epsilon^2}{4}\sum_{t=1}^{\tau-1}\frac{c(t+1)}{\alpha(t)\beta(t)^2} + \mathbb{E}[\psi] + L\epsilon\sum_{t=1}^{\tau}\frac{c(t)}{\beta(t)}. \quad (33)$$

Define $\gamma = \max_{t\in\{1,\tau-1\}}\frac{c(t+1)}{b(t)}$, $c_{\text{max}} = \max_{t\in[\tau]}c(t)$, and $\delta = \max_{\{t,t'\}\in\{1,\tau-1\}}|c(t) - c(t')|$. Then

$$\mathbb{E}[R|b_{\text{tot}}(\tau), c_{\text{tot}}(\tau)] \leq c_{\text{max}}[F(w(1)) - F(w^*) + \beta(\tau)h(w^*)] + \sum_{t=2}^{\tau}\delta F(w^*)$$

$$+ \sum_{t=1}^{\tau-1}\frac{KD\epsilon}{\beta(t)} + \frac{\gamma\sigma^2}{4}\sum_{t=1}^{\tau-1}\frac{1}{\alpha(t)} + \frac{K^2\epsilon^2 c_{\text{max}}}{4}\sum_{t=1}^{\tau-1}\frac{1}{\alpha(t)\beta(t)^2} + \mathbb{E}[\psi] + L\epsilon c_{\text{max}}\sum_{t=1}^{\tau}\frac{1}{\beta(t)} \quad (34)$$

In App. D, we bound $\sum_{t=1}^{\tau-1}\frac{1}{\alpha(t)}$ and $\sum_{t=1}^{\tau-1}\frac{1}{\alpha(t)\beta(t)^2}$ terms. Using them, we have

$$\mathbb{E}[R|b_{\text{tot}}(\tau), c_{\text{tot}}(\tau)] \leq c_{\text{max}}[F(w(1)) - F(w^*) + \beta(\tau)h(w^*)] + \delta\tau F(w^*) + 2KD\epsilon\sqrt{\mu\tau}$$

$$+ \frac{2\gamma\sigma^2\sqrt{\mu\tau}}{4} + \frac{3K^2\epsilon^2 c_{\text{max}}\mu^{3/2}}{4} + 2L\epsilon c_{\text{max}}\sqrt{\mu\tau} + \mathbb{E}[\psi] \quad (35)$$

$$\mathbb{E}[R|b_{\text{tot}}(\tau), c_{\text{tot}}(\tau)] \leq c_{\text{max}}[F(w(1)) - F(w^*) + \beta(\tau)h(w^*)] + \delta\tau F(w^*)$$

$$+ \frac{3K^2\epsilon^2 c_{\text{max}}\mu^{3/2}}{4} + \left(2KD\epsilon + \frac{\gamma\sigma^2}{2} + 2L\epsilon c_{\text{max}}\right)\sqrt{\mu\tau} + \mathbb{E}[\psi] \quad (36)$$

Now we bound $\mathbb{E}[\psi]$. Using $\delta = \max_{\{t,t'\}\in\{1,\tau-1\}}|c(t) - c(t')|$ in equation 32, we can write

$$\mathbb{E}[\psi] \leq \sum_{t=1}^{\tau-1}\delta\left((t-1)F(w^*) + \beta(t)h(w^*) + 2KD\epsilon\sqrt{\mu t}\right)$$

$$\leq \delta\left(F(w^*)\sum_{t=1}^{\tau-1}(t-1) + h(w^*)\sum_{t=1}^{\tau-1}\left(K + \sqrt{\frac{t}{\mu}}\right) + \sum_{t=1}^{\tau-1}2KD\epsilon\sqrt{\mu t}\right)$$

$$\leq \delta\left(\frac{\tau^2}{2}F(w^*) + h(w^*)\left(K + \sqrt{\frac{\tau}{\mu}}\right)\tau + 2KD\epsilon\sqrt{\mu\tau}\tau\right)$$

$$\leq \delta\left(\frac{\tau}{2}F(w^*) + h(w^*)\left(K + \sqrt{\frac{\tau}{\mu}}\right) + 2KD\epsilon\sqrt{\mu\tau}\right)\tau \quad (37)$$

By substituting equation 37 in equation 36

$$\mathbb{E}[R|b_{\text{tot}}(\tau), c_{\text{tot}}(\tau)] \leq c_{\text{max}}[F(w(1)) - F(w^*) + \beta(\tau)h(w^*)] + \delta\tau F(w^*)$$

$$+ \frac{3K^2\epsilon^2 c_{\text{max}}\mu^{3/2}}{4} + \left(2KD\epsilon + \frac{\gamma\sigma^2}{2} + 2L\epsilon c_{\text{max}}\right)\sqrt{\mu\tau}$$

$$+ \delta\left(\frac{\tau}{2}F(w^*) + h(w^*)\left(K + \sqrt{\frac{\tau}{\mu}}\right) + 2KD\epsilon\sqrt{\mu\tau}\right)\tau \quad (38)$$

By rearranging terms

$$\mathbb{E}[R|b_{\text{tot}}(\tau), c_{\text{tot}}(\tau)] \leq c_{\text{max}}[F(w(1)) - F(w^*) + \beta(\tau)h(w^*)] + \frac{3K^2\epsilon^2 c_{\text{max}}\mu^{3/2}}{4}$$

$$+ \delta\left(\left(1 + \frac{\tau}{2}\right)F(w^*) + h(w^*)\left(K + \sqrt{\frac{\tau}{\mu}}\right)\right)\tau$$

$$+ \left(2KD\epsilon + \frac{\gamma\sigma^2}{2} + 2L\epsilon c_{\text{max}} + 2\delta KD\epsilon\tau\right)\sqrt{\mu\tau} \quad (39)$$

Let $\mu = c_{\mathrm{avg}} = (1/\tau) \sum_{t=1}^{\tau} c(t)$, then from equation 15 $\mu\tau = m$ and we substitute

$$
\mathbb{E}[R|b_{\mathrm{tot}}(\tau), c_{\mathrm{tot}}(\tau)] \leq c_{\max}[F(w(1)) - F(w^*) + \beta(\tau)h(w^*)] + \frac{3K^2\epsilon^2 c_{\max}\mu^{3/2}}{4}
$$
$$
+ \delta \left( \left(1 + \frac{\tau}{2}\right) F(w^*) + h(w^*)\left(K + \tau^{1/2}c_{\mathrm{avg}}^{-1/2}\right)\right)\tau
$$
$$
+ \left(2KD\epsilon + \frac{\gamma\sigma^2}{2} + 2L\epsilon c_{\max} + 2\delta KD\epsilon\tau\right)\sqrt{m}. \quad (40)
$$

This completes the proof of Theorem 2.

## C    PROOF OF LEMMA 8

Note that $g(t)$ is calculated with respect to $w_i(t)$ by different nodes in equation 3. Let $\bar{g}(t)$ be the minibatch calculated with respect to $w(t)$ (given in equation 23) by all the nodes.

$$
\bar{g}(t) = \frac{1}{b(t)} \sum_{i=1}^{n} \sum_{s=1}^{b_i(t)} \nabla_w f(w(t), x_i(t, s)). \quad (41)
$$

Note that there are two types of errors in computing gradients. The first is common in any gradient based methods. That is, the gradient is calculated with respect to the function $f(w, x)$, which is based on the data $x$ instead of being a direct evaluation of $\nabla_w F(w)$. We denote this error as $q(t)$:

$$
q(t) = \bar{g}(t) - \nabla_w F(w(t)). \quad (42)
$$

The second error results from the fact that we use $g(t)$ instead of $\bar{g}(t)$. We denote this error as $r(t)$:

$$
r(t) = g(t) - \bar{g}(t). \quad (43)
$$

**Lemma 9** *The following four relations hold*

$$
\mathbb{E}[\langle q(t), w^* - w(t)\rangle] = 0,
$$

$$
\mathbb{E}[\langle r(t), w^* - w(t)\rangle] = \frac{KD\epsilon}{\beta(t)},
$$

$$
\mathbb{E}[\|q(t)\|^2] = \frac{\sigma^2}{b(t)},
$$

$$
\mathbb{E}[\|r(t)\|^2] = \frac{K^2\epsilon^2}{\beta(t)^2}.
$$

The proof of Lemma 9 is given in App. E. Let $l_t(w)$ be the first order approximation of $F(w)$ at $w(t)$:

$$
l_t(w) = F(w(t)) + \langle \nabla_w F(w(t)), w - w(t)\rangle. \quad (44)
$$

Let $\tilde{l}_t(w)$ be an approximation of $l_t(w)$ by replacing $\nabla_w F(w(t))$ with $g(t)$

$$
\tilde{l}_t(w) = F(w(t)) + \langle g(t), w - w(t)\rangle \quad (45)
$$
$$
= F(w(t)) + \langle \nabla_w F(w(t)), w - w(t)\rangle + \langle q(t), w - w(t)\rangle + \langle r(t), w - w(t)\rangle \quad (46)
$$
$$
= l_t(w) + \langle q(t), w - w(t)\rangle + \langle r(t), w - w(t)\rangle. \quad (47)
$$

Note that equation 46 follows since $g(t) = q(t) + r(t) + \nabla_w F(w(t))$. By using the smoothness of $F(w)$, we can write

$$
F(w(t+1)) \leq l_t(w(t+1)) + \frac{K}{2}\|w(t+1) - w(t)\|^2
$$
$$
= \tilde{l}_t(w(t+1)) - \langle q(t), w - w(t)\rangle - \langle r(t), w - w(t)\rangle + \frac{K}{2}\|w(t+1) - w(t)\|^2
$$
$$
= \tilde{l}_t(w) + \|q(t)\|\|w - w(t)\| + \|r(t)\|\|w - w(t)\| + \frac{K}{2}\|w(t+1) - w(t)\|^2. \quad (48)
$$

The last step is due to the Cauchy-Schwarz inequality. Let $\alpha(t) = \beta(t) - K$. We add and subtract $\alpha(t)\|w(t+1) - w(t)\|^2/2$ to find

$$F(w(t+1)) \leq \tilde{l}_t(w(t+1)) + \|q(t)\|\|w - w(t)\| - \frac{\alpha(t)}{4}\|w(t+1) - w(t)\|^2 + \|r(t)\|\|w - w(t)\|$$
$$- \frac{\alpha(t)}{4}\|w(t+1) - w(t)\|^2 + \frac{K + \alpha(t)}{2}\|w(t+1) - w(t)\|^2.$$

Note that

$$\|q(t)\|\|w - w(t)\| - \frac{\alpha(t)}{4}\|w(t+1) - w(t)\|^2$$
$$= \frac{\|q(t)\|^2}{4\alpha(t)} - \left[\frac{\|q(t)\|}{\sqrt{4\alpha(t)}} - \sqrt{\frac{a(t)}{4}}\|w(t+1) - w(t)\|\right]^2 \leq \frac{\|q(t)\|^2}{4\alpha(t)}. \tag{49}$$

Similarly, we have that

$$\|r(t)\|\|w - w(t)\| - \frac{\alpha(t)}{4}\|w(t+1) - w(t)\|^2 \leq \frac{\|r(t)\|^2}{4\alpha(t)}. \tag{50}$$

Using equation 49, equation 50, and $\beta(t) = K + \alpha(t)$ in equation 48 we have

$$F(w(t+1)) \leq \tilde{l}_t(w(t+1)) + \frac{\beta(t)}{2}\|w(t+1) - w(t)\|^2 + \frac{\|q(t)\|^2}{4\alpha(t)} + \frac{\|r(t)\|^2}{4\alpha(t)} \tag{51}$$

The following Lemma gives a relation between $w(t)$ and $\tilde{l}_t(w(t))$

**Lemma 10** *The optimization stated in equation 23 is equivalent to*

$$w(t) = \arg\min_{w \in W}\left\{\sum_{t'=1}^{t-1}\tilde{l}_{t'}(w) + \beta(t)h(w)\right\}. \tag{52}$$

By using the result (Dekel et al., 2012, Lemma 8), we have

$$\frac{\beta(t)}{2}\|w(t+1) - w(t)\|^2 \leq \sum_{t'=1}^{t-1}\tilde{l}_{t'}(w(t+1)) + (\beta(t))h(w(t+1))$$
$$- \sum_{t'=1}^{t-1}\tilde{l}_{t'}(w(t)) + (K + \beta(t))h(w(t)). \tag{53}$$

Use equation 51 in equation 53 and substituting in $\beta(t) = K + \alpha(t)$ we get

$$F(w(t+1)) \leq l_t(w(t+1)) + \sum_{t'=1}^{t-1}\tilde{l}_{t'}(w(t+1)) - \sum_{t'=1}^{t-1}\tilde{l}_{t'}(w(t)) + (K + \alpha(t))h(w(t+1))$$
$$- (K + \alpha(t))h(w(t)) + \frac{\|q(t)\|^2 + \|r(t)\|^2}{4\alpha(t)}$$
$$\leq \sum_{t'=1}^{t}\tilde{l}_{t'}(w(t+1)) - \sum_{t'=1}^{t-1}\tilde{l}_{t'}(w(t)) + (K + \alpha(t+1))h(w(t+1))$$
$$- (K + \alpha(t))h(w(t)) + \frac{\|q(t)\|^2 + \|r(t)\|^2}{4\alpha(t)}, \tag{54}$$

where equation 54 is due to the fact that $\alpha(t+1) \geq \alpha(t)$. Now, we use $\beta(t) = K + \alpha(t)$, multiply by $c(t+1)$ and rewrite

$$
\begin{aligned}
c(t+1)F(w(t+1)) \leq &\sum_{t'=1}^{t} \tilde{c}(t+1)l_{t'}(w(t+1)) - \sum_{t'=1}^{t-1} \tilde{c}(t)l_{t'}(w(t)) + c(t+1)\beta(t+1)h(w(t+1)) \\
&- c(t)\beta(t)h(w(t)) + c(t+1)\frac{\|q(t)\|^2 + \|r(t)\|^2}{4\alpha(t)} \\
&- (c(t+1) - c(t))\left[\sum_{t'=1}^{t-1} \tilde{l}_{t'}(w(t)) + \beta(t)h(w(t))\right]. \quad (55)
\end{aligned}
$$

Summing from $t = 1$ to $\tau - 1$ we get

$$
\begin{aligned}
\sum_{t=2}^{\tau} c(t)F(w(t)) \leq &\sum_{t=1}^{\tau-1} c(\tau)\tilde{l}_t(w(\tau)) + c(\tau)\beta(\tau)h(w(\tau)) + \sum_{t=1}^{\tau-1} c(t+1)\frac{\|q(t)\|^2 + \|r(t)\|^2}{4\alpha(t)} \\
&+ \sum_{t=1}^{\tau-1}(c(t) - c(t+1))\left[\sum_{t'=1}^{t-1} \tilde{l}_{t'}(w(t)) + \beta(t)h(w(t))\right].
\end{aligned}
$$

Let $\psi$ be the last two terms, i.e.,

$$
\psi = \sum_{t=1}^{\tau-1}(c(t) - c(t+1))\left[\sum_{t'=1}^{t-1} \tilde{l}_{t'}(w(t)) + \beta(t)h(w(t))\right]. \quad (56)
$$

Then, using Lemma 10

$$
\sum_{t=2}^{\tau} c(t)F(w(t)) \leq \sum_{t=1}^{\tau-1} c(\tau)\tilde{l}_t(w^*) + c(\tau)\beta(\tau)h(w^*) + \sum_{t=1}^{\tau-1} c(t+1)\frac{\|q(t)\|^2 + \|r(t)\|^2}{4\alpha(t)} + \psi.
$$

By substituting in equation 45 we continue

$$
\begin{aligned}
\sum_{t=2}^{\tau} c(t)F(w(t)) \leq &\sum_{t=1}^{\tau-1} c(\tau)l_t(w^*) + \sum_{t=1}^{\tau-1}\langle q(t), w - w(t)\rangle + \langle r(t), w - w(t)\rangle \\
&+ c(\tau)\beta(\tau)h(w^*) + \sum_{t=1}^{\tau-1} c(t+1)\frac{\|q(t)\|^2 + \|r(t)\|^2}{4\alpha(t)} + \psi \\
\leq &(\tau - 1)c(\tau)F(w^*) + \sum_{t=1}^{\tau-1}\langle q(t), w - w(t)\rangle + \langle r(t), w - w(t)\rangle \\
&+ c(\tau)\beta(\tau)h(w^*) + \sum_{t=1}^{\tau-1} c(t+1)\frac{\|q(t)\|^2 + \|r(t)\|^2}{4\alpha(t)} + \psi \quad (57)
\end{aligned}
$$

where equation 57 is due to convexity of $F(w)$, i.e., $\sum_{t=1}^{\tau-1} l_t(w^*) \leq (\tau - 1)F(w^*)$. Adding and subtracting terms we find that

$$
\begin{aligned}
\sum_{t=1}^{\tau} c(t)[F(w(t)) - F(w^*)] \leq &c(1)[F(w(1)) - F(w^*)] + \sum_{t=2}^{\tau}(c(\tau) - c(t))F(w^*) \\
&+ \sum_{t=1}^{\tau-1}\langle q(t), w - w(t)\rangle + \langle r(t), w - w(t)\rangle \\
&+ c(\tau)\beta(\tau)h(w^*) + \sum_{t=1}^{\tau-1} c(t+1)\frac{\|q(t)\|^2 + \|r(t)\|^2}{4\alpha(t)} + \psi
\end{aligned}
$$

Taking the expectation with respect to $X(\tau - 1)$

$$\mathbb{E}\left[\sum_{t=1}^{\tau} c(t)[F(w(t)) - F(w^*)]\right] \le c(1)[F(w(1)) - F(w^*)] + \sum_{t=2}^{\tau}(c(\tau) - c(t))F(w^*)$$

$$+ c(\tau)\beta(\tau)h(w^*) + \sum_{t=1}^{\tau-1}\mathbb{E}[\langle q(t), w - w(t)\rangle] + \mathbb{E}[\langle r(t), w - w(t)\rangle]$$

$$+ \sum_{t=1}^{\tau-1} c(t+1)\frac{\mathbb{E}[\|q(t)\|^2] + [\|r(t)\|^2]}{4\alpha(t)} + \mathbb{E}[\psi]$$

We use the bounds in Lemma 9 to get

$$\mathbb{E}\left[\sum_{t=1}^{\tau} c(t)[F(w(t)) - F(w^*)]\right] \le c(1)[F(w(1)) - F(w^*)] + \sum_{t=2}^{\tau}(c(\tau) - c(t))F(w^*)$$

$$+ c(\tau)\beta(\tau)h(w^*) + \sum_{t=1}^{\tau-1}\frac{KD\epsilon}{\beta(t)} + \sum_{t=1}^{\tau-1}\frac{c(t+1)}{4\alpha(t)}\left(\frac{\sigma^2}{b(t)} + \frac{K^2\epsilon^2}{\beta(t)^2}\right) + \mathbb{E}[\psi].$$

We rewrite by rearranging terms

$$\mathbb{E}\left[\sum_{t=1}^{\tau} c(t)[F(w(t)) - F(w^*)]\right] \le c(1)[F(w(1)) - F(w^*)] + \sum_{t=2}^{\tau}(c(\tau) - c(t))F(w^*)$$

$$+ c(\tau)\beta(\tau)h(w^*) + \sum_{t=1}^{\tau-1}\frac{KD\epsilon}{\beta(t)} + \sum_{t=1}^{\tau-1}\frac{c(t+1)\sigma^2}{4b(t)\alpha(t)} + \frac{K^2\epsilon^2}{4}\sum_{t=1}^{\tau-1}\frac{c(t+1)}{\alpha(t)\beta(t)^2} + \mathbb{E}[\psi] \quad (58)$$

Now we bound $\mathbb{E}[\psi]$. From equation 56 we find

$$\psi = \sum_{t=1}^{\tau-1}(c(t) - c(t+1))\left(\sum_{t'=1}^{t-1}\tilde{l}_{t'}(w(t)) + \beta(t)h(w(t))\right)$$

$$\le \sum_{t=1}^{\tau-1}(c(t) - c(t+1))\left(\sum_{t'=1}^{t-1}\tilde{l}_{t'}(w^*) + \beta(t)h(w^*)\right) \quad (59)$$

$$= \sum_{t=1}^{\tau-1}(c(t) - c(t+1))\left(\sum_{t'=1}^{t-1}(l_{t'}(w^*) + \langle q(t'), w^* - w(t')\rangle + \langle r(t'), w^* - w(t')\rangle) + \beta(t)h(w^*)\right) \quad (60)$$

$$\le \sum_{t=1}^{\tau-1}(c(t) - c(t+1))\left((t-1)F(w^*) + \beta(t)h(w^*) + \sum_{t'=1}^{t-1}\langle q(t'), w^* - w(t')\rangle + \langle r(t'), w^* - w(t')\rangle\right), \quad (61)$$

where equation 59 is due to Lemma 10, equation 60 is simple substitution of equation 45, and the last step is due to convexity of $F(w)$. Now, we take the expectation over data samples $X(\tau - 1)$

$\mathbb{E}[\psi]$

$$\le \sum_{t=1}^{\tau-1}(c(t) - c(t+1))\left((t-1)F(w^*) + \beta(t)h(w^*) + \sum_{t'=1}^{t-1}\mathbb{E}[\langle q(t'), w^* - w(t')\rangle] + \mathbb{E}[\langle r(t'), w^* - w(t')\rangle]\right)$$

$$\le \sum_{t=1}^{\tau-1}(c(t) - c(t+1))\left((t-1)F(w^*) + \beta(t)h(w^*) + \sum_{t'=1}^{t-1}\frac{KD\epsilon}{\beta(t')}\right) \quad (62)$$

$$\le \sum_{t=1}^{\tau-1}(c(t) - c(t+1))\left((t-1)F(w^*) + \beta(t)h(w^*) + 2KD\epsilon\sqrt{\mu t}\right) \quad (63)$$

where Lemma 9 is used in equation 62 and the last step is due to equation 64. This completes the proof of Lemma 8.

## D    PROOF OF BOUNDS USED IN APP. F

We know $\beta(t) = K + \alpha(t)$. Let $\alpha(t) = \sqrt{\frac{t}{\mu}}$. Then, we have

$$\sum_{t=1}^{\tau-1} \frac{1}{\beta(t)} \leq 2\sqrt{\mu\tau}. \tag{64}$$

Similarly,

$$\begin{aligned}
\sum_{t=1}^{\tau-1} \frac{1}{\alpha(t)\beta(t)^2} &= \sum_{t=1}^{\tau-1} \frac{1}{\alpha(t)(K+\alpha(t)^2)} \\
&\leq \sum_{t=1}^{\tau-1} \frac{1}{\alpha(t)^3} \\
&= \mu^{3/2} \sum_{t=1}^{\tau-1} t^{-3/2} \\
&\leq \mu^{3/2} \left(1 + \int_1^\tau t^{-3/2} dt\right) \\
&\leq 3\mu^{3/2}. \tag{65}
\end{aligned}$$

## E    PROOF OF LEMMA 9

Note that the expectation with respect to $x_s(t)$

$$\mathbb{E}[f(w(t), x_s(t))] = \mathbb{E}[F(w(t))]. \tag{66}$$

Also we use the fact that gradient and expectation operators commutes

$$\mathbb{E}[\langle \nabla f(w(t)), x_s(t), w(t) \rangle] = \mathbb{E}[\langle F(w(t)), w(t) \rangle]. \tag{67}$$

Bounding $\mathbb{E}[\langle q(t), w^* - w(t) \rangle]$ and $\mathbb{E}[\|q(t)\|^2]$ follows the same approach as in (Dekel et al., 2012, Appendix A.1) or Tsianos & Rabbat (2016). Now, we find $\mathbb{E}[\langle r(t), w^* - w(t) \rangle]$

$$\mathbb{E}[\langle r(t), w^* - w(t) \rangle]$$

$$= \mathbb{E}\left[\left\langle \frac{1}{b(t)} \sum_{i=1}^n \sum_{s=1}^{b_i(t)} \nabla_w f(w_i(t), x_i(t, s)) - \frac{1}{b(t)} \sum_{i=1}^n \sum_{s=1}^{b_i(t)} \nabla_w f(w(t), x_i(t, s)), w^* - w(t) \right\rangle\right]$$

$$= \mathbb{E}\left[\left\langle \frac{1}{b(t)} \sum_{i=1}^n \sum_{s=1}^{b_i(t)} \nabla_w F(w_i(t)) - \frac{1}{b(t)} \sum_{i=1}^n \sum_{s=1}^{b_i(t)} \nabla_w F(w(t)), w^* - w(t) \right\rangle\right]$$

$$= \frac{1}{b(t)} \sum_{i=1}^n b_i(t) \mathbb{E}\left[\langle \nabla_w F(w_i(t)) - \nabla_w F(w(t)), w^* - w(t) \rangle\right]$$

$$\leq \frac{1}{b(t)} \sum_{i=1}^n b_i(t) \mathbb{E}\left[\|\nabla_w F(w_i(t)) - \nabla_w F(w(t))\| \|w^* - w(t)\|\right] \tag{68}$$

$$\leq \frac{1}{b(t)} \sum_{i=1}^n b_i(t) \mathbb{E}\left[K\|w_i(t) - w(t)\| D\right] \tag{69}$$

where equation 68 is due to the Cauchy-Schwarz inequality and equation 69 due to equation 9 and $D = \max_{w,u \in W} \|w - u\|$. Using equation 24

$$\begin{aligned}
\mathbb{E}[\langle r(t), w^* - w(t) \rangle] &\leq \frac{1}{b(t)} \sum_{i=1}^n \frac{b_i(t) K D \epsilon}{\beta(t)} \\
&= \frac{K D \epsilon}{\beta(t)}. \tag{70}
\end{aligned}$$

Now we find $\mathbb{E}[\|r(t)\|^2]$.

$$
\begin{aligned}
\mathbb{E}[\|r(t)\|^2] &= \mathbb{E}\left[\left\|\frac{1}{b(t)}\sum_{i=1}^{n}\sum_{s=1}^{b_i(t)}\nabla_w f(w_i(t), x_i(t,s)) - \nabla_w f(w(t), x_i(t,s))\right\|^2\right] \\
&\leq \mathbb{E}\left[\left(\frac{1}{b(t)}\sum_{i=1}^{n}\sum_{s=1}^{b_i(t)}\|\nabla_w f(w_i(t), x_i(t,s)) - \nabla_w f(w(t), x_i(t,s))\|\right)^2\right] \\
&\leq \mathbb{E}\left[\left(\frac{1}{b(t)}\sum_{i=1}^{n}\sum_{s=1}^{b_i(t)}K\|w_i(t) - w(t)\|\right)^2\right] \\
&\leq \left(\frac{1}{b(t)}\sum_{i=1}^{n}\sum_{s=1}^{b_i(t)}\frac{K\epsilon}{\beta(t)}\right)^2 \\
&= \frac{K^2\epsilon^2}{\beta(t)^2}.
\end{aligned}
\tag{71}
$$

## F    PROOF OF THEOREM 4

By definition

$$
c(t) = \sum_{i=1}^{n} c_i(t)
\tag{72}
$$

where $c_i(t)$ is the total number of gradients computed at the node $i$ in the $t$-th epoch. We assume $c_i(t)$ is independent across network and is independent and identically distributed according to some processing time distribution $p$ across epochs. Let $\bar{c} = \mathbb{E}_p[c(t)]$ and let $1/\hat{b} = \mathbb{E}_p[1/b(t)]$. From Lemma 8 we have that

$$
\begin{aligned}
\sum_{t=1}^{\tau}c(t)\mathbb{E}\left[F(w(t)) - F(w^*)\right] &\leq c(1)[F(w(1)) - F(w^*)] + \sum_{t=2}^{\tau}(c(\tau) - c(t))F(w^*) \\
&\quad + c(\tau)\beta(\tau)h(w^*) + \sum_{t=1}^{\tau-1}\frac{KD\epsilon}{\beta(t)} \\
&\quad + \sum_{t=1}^{\tau-1}\frac{c(t+1)\sigma^2}{4b(t)\alpha(t)} + \frac{K^2\epsilon^2}{4}\sum_{t=1}^{\tau-1}\frac{c(t+1)}{\alpha(t)\beta(t)^2} + \mathbb{E}[\psi].
\end{aligned}
\tag{73}
$$

Let $\alpha(t) = \sqrt{t/\bar{c}}$. Now take expectation over the $c(t)$ to get

$$
\begin{aligned}
\mathbb{E}_p\left[\sum_{t=1}^{\tau}c(t)\mathbb{E}\left[F(w(t)) - F(w^*)\right]\right] &\leq \bar{c}[F(w(1)) - F(w^*)] + \bar{c}\beta(\tau)h(w^*) + \sum_{t=1}^{\tau-1}\frac{KD\epsilon}{\beta(t)} \\
&\quad + \sum_{t=1}^{\tau-1}\mathbb{E}_p\left[\frac{c(t+1)}{4\alpha(t)}\left(\frac{\sigma^2}{b(t)} + \frac{K^2\epsilon^2}{\beta(t)^2}\right)\right] + \mathbb{E}_p[\mathbb{E}[\psi]] \\
&= \bar{c}[F(w(1)) - F(w^*)] + \bar{c}\beta(\tau)h(w(\tau)) + \sum_{t=1}^{\tau-1}\frac{KD\epsilon}{\beta(t)} \\
&\quad + \sum_{t=1}^{\tau-1}\frac{\bar{c}}{4\alpha(t)}\left(\frac{\sigma^2}{\hat{b}} + \frac{K^2\epsilon^2}{\beta(t)^2}\right).
\end{aligned}
\tag{74}
$$

The last step is due to the fact that $c(t+1)$ and $b(t)$ are independent since these are in two different epochs. Further $\mathbb{E}_p[\mathbb{E}[\psi|c(t)]] = 0$. After further simplification through the use of Appendix D, we get

$$\mathbb{E}_p\left[\sum_{t=1}^{\tau} c(t)\mathbb{E}\left[F(w(t)) - F(w^*)\right]\right] \leq \bar{c}[F(w(1)) - F(w^*) + \bar{\beta}(\tau)h(w(\tau))] + 2KD\epsilon\sqrt{\bar{c}\tau}$$
$$+ \frac{\bar{c}\sigma^2\sqrt{\bar{c}\tau}}{2\hat{b}} + \frac{3K^2\epsilon^2\bar{c}^{5/2}}{4}. \tag{75}$$

Taking the expectation over $c(t)$ in equation 30, we have

$$\mathbb{E}_p[\mathbb{E}[R|b_{\text{tot}}(\tau), c_{\text{tot}}(\tau)]] \leq \mathbb{E}_p\left[\sum_{t=1}^{\tau} c(t)\mathbb{E}\left[F(w(t)) - F(w^*)\right]\right] + L\epsilon\sum_{t=1}^{\tau}\frac{\bar{c}}{\beta(t)}$$
$$\leq \bar{c}[F(w(1)) - F(w^*) + \bar{\beta}(\tau)h(w(\tau))] + 2KD\epsilon\sqrt{\bar{c}\tau}$$
$$+ \frac{\bar{c}\sigma^2\sqrt{\bar{c}\tau}}{2\hat{b}} + \frac{3K^2\epsilon^2\bar{c}^{5/2}}{4} + 2L\epsilon\bar{c}\sqrt{\bar{c}\tau}. \tag{76}$$

By definition

$$m = \sum_{t=1}^{\tau} c(t). \tag{77}$$

Then $\bar{m} = \mathbb{E}_p = \bar{c}\tau$. By substituting $\bar{m}$ and rearranging we find that

$$\mathbb{E}_p[\mathbb{E}[R|b_{\text{tot}}(\tau), c_{\text{tot}}(\tau)]] \leq \bar{c}[F(w(1)) - F(w^*) + \bar{\beta}(\tau)h(w(\tau))] + \frac{3K^2\epsilon^2\bar{c}^{5/2}}{4}$$
$$+ \left(2KD\epsilon + \frac{\bar{c}\sigma^2}{2\hat{b}} + 2L\epsilon\bar{c}\right)\sqrt{\bar{m}} \tag{78}$$

## G PROOF OF THEOREM 7

*Proof:* Consider an FMB method in which each node computes $b/n$ gradients per epoch, with $T_i(t)$ denoting the time taken to complete the job.

Also consider AMB with a fixed epoch duration of $T$. The number of gradient computations completed by the $i$-th node in the $t$-th epoch is

$$b_i(t) = \left\lfloor \frac{bT}{nT_i(t)} \right\rfloor \geq \frac{bT}{nT_i(t)} - 1. \tag{79}$$

Therefore, the minibatch size $b(t)$ computed in AMB in the $t$-th epoch is

$$b(t) = \sum_{i=1}^{n} b_i(t) \geq \sum_{i=1}^{n} \frac{bT}{nT_i(t)} - n = \frac{bT}{n}\sum_{i=1}^{n}\frac{1}{T_i(t)} - n. \tag{80}$$

Taking the expectation over the distribution of $T_i(t)$ in (80), and applying Jensen's inequality, we find that

$$\mathbb{E}_p[b(t)] \geq \frac{bT}{n}\sum_{i=1}^{n}\mathbb{E}_p\left[\frac{1}{T_i(t)}\right] - n \geq \frac{bT}{n}\sum_{i=1}^{n}\frac{1}{\mathbb{E}_p[T_i(t)]} - n = bT\mu^{-1} - n.$$

where $E_p[T_i(t)] = mu$. Fixing the computing time to $T = (1 + n/b)\mu$ we find that $\mathbb{E}_p[b(t)] \geq b$, i.e., the expected minibatch of AMB is at least as large as the minibatch size $b$ used in the FMB.

The expected computing time for $\tau$ epochs in our approach is

$$S_A = \tau T = \tau(1 + n/b)\mu. \tag{81}$$

In contrast, in the FMB approach the finishing time of the $t$th epoch is $\max_{i \in [n]} T_i(t)$. Using the result of Arnold & Groeneveld (1979); Bertsimas et al. (2006) we find that

$$\mathbb{E}_p[\max_{i \in [n]} T_i(t)] \leq \mu + \sigma \sqrt{n-1}, \tag{82}$$

where $\sigma$ is the standard deviation of $T_i(t)$. Thus $\tau$ epochs takes expected time

$$S_F = \tau \mathbb{E}_p[\max_{i \in [n]} T_i(t)] \leq \tau \left( \mu + \sigma \sqrt{n-1} \right) \tag{83}$$

Taking the ratio of the two finishing times we find that

$$\frac{S_F}{S_A} \leq \frac{\mu + \sigma \sqrt{n-1}}{(1 + n/b)\mu} = \left( 1 + \frac{\sigma}{\mu} \sqrt{n-1} \right) (1 + n/b)^{-1}. \tag{84}$$

For parallelization to be meaningful, the minibatch size should be much larger than number of nodes and hence $b \gg n$. This means $(1 + n/b) \approx 1$ for any system of interest. Thus,

$$S_F \leq \left( 1 + \frac{\sigma}{\mu} \sqrt{n-1} \right) S_A, \tag{85}$$

This completes the proof of Theorem 7.

## H  SHIFTED EXPONENTIAL DISTRIBUTION

The shifted exponential distribution is given by

$$p_{T_i(t)}(z) = \lambda \exp \left( -\lambda(z - \zeta) \right), \ \ \forall z \geq \zeta \tag{86}$$

where $\lambda \geq 0$ and $\zeta \geq 0$. The shifted exponential distribution models a minimum time ($\zeta$) to complete a job, and a memoryless balance of processing time thereafter. The $\lambda$ parameter dictates the average processing speed, with larger $\lambda$ indicating faster processing. The expected finishing time is $\mathbb{E}_p[T_i(t)] = \lambda^{-1} + \zeta$. Therefore,

$$S_A = \tau T = \tau(1 + n/b)(\lambda^{-1} + \zeta). \tag{87}$$

By using order statistics, we can find

$$\mathbb{E}_p[\max_{i \in [n]} T_i(t)] = \lambda^{-1} \log(n) + \zeta, \tag{88}$$

and thus $\tau$ epochs takes expected time

$$S_F = \tau \left( \lambda^{-1} \log(n) + \zeta \right) \tag{89}$$

Taking the ratio of the two finishing times we find that

$$\frac{S_F}{S_A} = \frac{\left( \lambda^{-1} \log(n) + \zeta \right)}{(1 + n/b)(\lambda^{-1} + \zeta)}. \tag{90}$$

For parallelization to be meaningful we must have much more data than nodes and hence $b \gg n$. This means that the first factor in the denominator will be approximately equal to one for any system of interest. Therefore, in the large $n$ regime,

$$\lim_{n \to \infty} S_F = \frac{\log(n)}{1 + \lambda \zeta} S_A, \tag{91}$$

which is order-$\log(n)$ since the product $\lambda \zeta$ is fixed.

## I  SUPPORTING DISCUSSION OF NUMERICAL RESULTS OF MAIN PAPER AND ADDITIONAL EXPERIMENTS

In this section, we present additional details regarding the numerical results of Section 6 of the main paper as well as some new results. In Appendix I.1, we detail the network used in Section 6 and, for a point of comparison, implement the same computations in a master-worker network

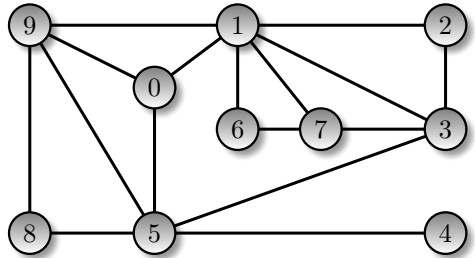

Figure 2: The topology of the network used in our experiments on distributed optimization.

topology. In Appendix I.2, we model the compute times of the nodes as shifted exponential random variables and, under this model, present results contrasting AMB and FMB performance for the linear regression problem. In Appendix I.3 we present an experimental methodology for simulating a wide variety of straggler distributions in EC2. By running background jobs on some of the EC2 nodes we slow the foreground job of interest, thereby simulating a heavily-loaded straggler node. Finally, in Appendix I.4, we present another experiment in which we also induce stragglers by forcing the nodes to make random pauses between two consecutive gradient calculations. We present numerical results for both settings as well, demonstrating the even greater advantage of AMB versus FMB when compared to the results presented in Section 6.

### I.1 SUPPORTING DETAILS FOR RESULTS OF SECTION 6 AND COMPARATIVE RESULTS FOR HUB-AND-SPOKE TOPOLOGY

As there was not space in the main text, in Figure 2 we diagram the connectivity of the distributed computation network used in Section 6. The second largest eigenvalue of the $P$ matrix corresponding to this network, which controls the speed of consensus, is 0.888.

In Section 6, we presented results for distributed logistic regression in the network depicted in Figure 2. Another network topology of great interest is the hub-and-spoke topology wherein a central master node is directly connected to a number of worker nodes, and worker nodes are only indirectly connected via the master. We also ran the MNIST logistic regression experiments for this topology. In our experiments there were 20 nodes total, 19 workers and one master. As in Sec.6 we used t2.micro instances and ami-62b11202 to launch the instances. We set the total batch size used in FMB to be $b = 3990$ so, with $n = 19$ worker each worker calculated $b/n = 210$ gradients per batch. Working with this per-worker batch size, we found the average EC2 compute time per batch to be 3 sec. Therefore, we used a compute time of $T = 3$ sec. in the AMB scheme while the communication time of $T_c = 1$ sec. Figure 3 plots the logistical error versus wall clock time for both AMB and FMB in the master-worker (i.e., hub-and-spoke) topology. We see that the workers implementing AMB far outperform those implementing FMB.

### I.2 MODELING STRAGGLERS USING A SHIFTED EXPONENTIAL DISTRIBUTION

In this section, we model the speed of each worker probabilistically. Let $T_i(t)$ denote the time taken by worker $i$ to calculate a total of 600 gradients in the $t$-th epoch. We assume $T_i(t)$ follows a shifted exponential distribution and is independent and identically distributed across nodes (indexed by $i$) and across computing epochs (indexed by $t$). The probability density function of the shifted exponential is $p_{T_i(t)}(z) = \lambda e^{-\lambda(z-\zeta)}$. The mean of this distribution is $\mu = \zeta + \lambda^{-1}$ and its variance is $\lambda^{-2}$. Conditioned on $T_i(t)$ we assume that worker $i$ makes linear progress through the dataset. In other words, worker $i$ takes $kT_i(t)/600$ seconds to calculate $k$ gradients. (Note that our model allows $k$ to exceed 600.) In the simulation results we present we choose $\lambda = 2/3$ and $\zeta = 1$. In the AMB scheme, node $i$ computes $b_i(t) = 600T/T_i(t)$ gradients in epoch $t$ where $T$ is the fixed computing time allocated. To ensure a fair comparison between FMB and AMB, $T$ is chosen according to Thm. 7. This means that $\mathbb{E}[b(t)] \geq b$ where $b(t) = \sum_i b_i(t)$ and $b$ is the fixed minibatch size used by FMB. Based on our parameter choices, $T = (1 + n/b)\mu = (1 + n/b)(\lambda^{-1} + \zeta) = 2.5$.

Figure 4 plots the average error rate of the linear regression problem versus wall clock time for both FMB and AMB assuming a distributed computation network depicted in Figure 2. In these results, we

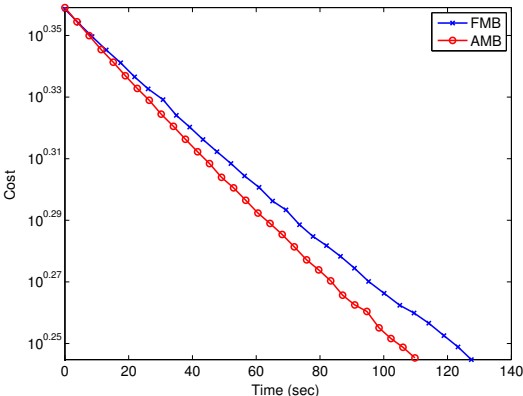

Figure 3: MNIST logistic regression training results for AMB and FMB operating in the hub-and-spoke topology.

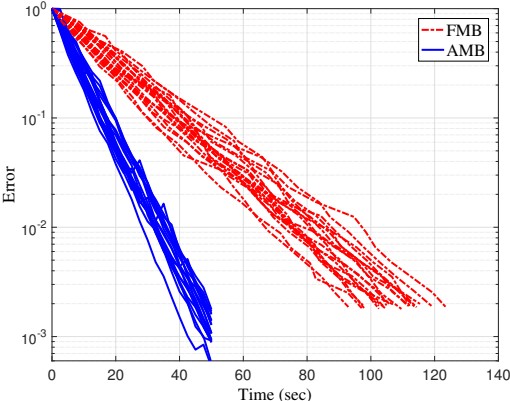

Figure 4: Linear regression error of AMB and FMB operating in the fully distributed topology of Figure 2 for 20 sample paths of $\{T_i(t))\}$ generated according the the shifted exponential distribution. We plot error versus wall clock time where there are five rounds of consensus.

generate 20 sample paths; each sample path is a set $\{T_i(t)\}$ for $i \in \{1, \ldots, 20\}$ and $t \in \{1, \ldots 20\}$. At the end of each of the 20 computing epoch we conduct $r = 5$ rounds of consensus. As can be observed in Fig. 4, for all 20 sample paths AMB outperforms FMB. One can also observe that there for neither scheme is there much variance in performance across sample paths; there is a bit more for FMB than for AMB. Due to this small variability, in the rest of this discussion we pick a single sample path to plot results for.

Figures 5a and 5b help us understand the performance impact of imperfect consensus on both AMB and on FMB. In each we plot the consensus error for $r = 5$ rounds of consensus and perfect consensus ($r = \infty$). In Fig. 5a we plot the error versus number of computing epochs while in Figure 5b we plot it versus wall clock time. In the former there is very little difference between AMB and FMB. This is due to the fact that we have set the computation times so that the expected AMB batch size equals the fixed FMB batch size. On the other hand, there is a large performance gap between the schemes when plotted versus wall clock time. It is thus in terms of real time (not epoch count) where AMB strongly outperforms FMB. In particular, AMB reaches an error rate of $10^{-3}$ in less than half the time that it takes FMB (2.24 time faster, to be exact).

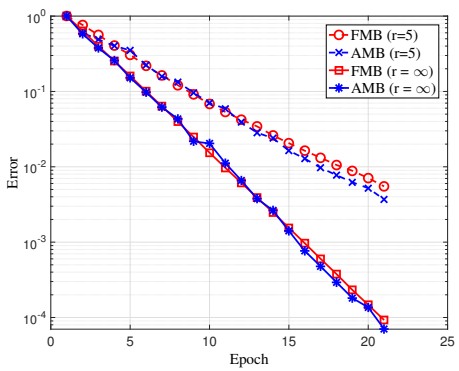 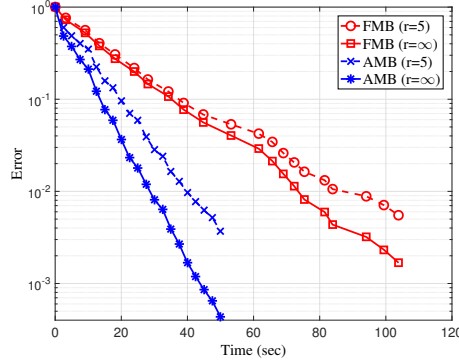

(a) Effect of imperfect consensus versus epoch.  (b) Effect of imperfect consensus versus time.

Figure 5: The effect of imperfect consensus on AMB and FMB.

### I.3 PERFORMANCE WITH INDUCED STRAGGLERS ON EC2

In this section, we introduce a new experimental methodology for studying the effect of stragglers. In these experiments we *induce* stragglers amongst our EC2 micro.t2 instances by running background jobs. In our experiments, there were 10 compute nodes interconnected according to the topology of Figure 2. The 10 worker nodes were partitioned into three groups. In the first group we run two background jobs that "interfere" with the foreground (AMB or FMB) job. The background jobs we used were matrix multiplication jobs that were continuously performed during the experiment. This first group will contain the "bad" straggler nodes. In the second group we run a single background job. These will be the intermediate stragglers. In the third group we do not run background jobs. These will be the non-stragglers. In our experiments, there are three bad stragglers (workers 1, 2, and 3), two intermediate stragglers (workers 4 and 5), and five non-stragglers (workers 6-10).

We first launch the background jobs in groups one and two. We then launch the FMB jobs on all nodes at once. By simultaneously running the background jobs and FMB, the resources of nodes in the first two groups are shared across multiple tasks resulting in an overall slowdown in their computing. The slowdown can be clearly observed in Figure 6a which depicts the histogram of the FMB compute times. The count ("frequency") is the number of jobs (fixed mini batches) completed as a function of the time it took to complete the job. The third (fast) group is on the left, clustered around 10 seconds per batch, while the other two groups are clustered at roughly 20 and 30 seconds. Figure 6b depicts the same experiment as performed with AMB: first launching the background jobs, and then launching AMB in parallel on all nodes. In this scenario compute time is fixed, so the histogram plots the number of completed batches completed as a function of batch size. In the AMB experiments the bad straggler nodes appear in the first cluster (centered around batch size of 230) while the faster nodes appear in the clusters to the right. In the FMB histogram per-worker batch size was fixed to 585 while in the AMB histograms the compute time was fixed to 12 sec.

We observe that these empirical results confirm the conditionally deterministic aspects of our statistical model of Appendix I.2. This was the portion of the model wherein we assumed that nodes make linear progress conditioned on the time it takes to compute one match. In Figure 6a, we observe it takes the non-straggler nodes about 10 seconds to complete one fixed-sized minibatch. It takes the intermediate nodes about twice as long. Turning to the AMB plots we observe that, indeed, the intermediate stragglers nodes complete only about 50% of the work that the non-straggler nodes do in the fixed amount of time. Hence this "linear progress" aspect of our model is confirmed experimentally.

Figure 7 illustrates the performance of AMB and FMB on the MNIST regression problem in the setting of EC2 with induced stragglers. As can be observed by comparing these results to those presented in Figure 1b of Section 6, the speedup now effected by AMB over FMB is far larger. While in Figure 1b the AMB was about 50% faster than FMB it is now about twice as fast. While previously AMB effect a reduction of 30% in the time it took FMB to hit a target error rate, the reduction now

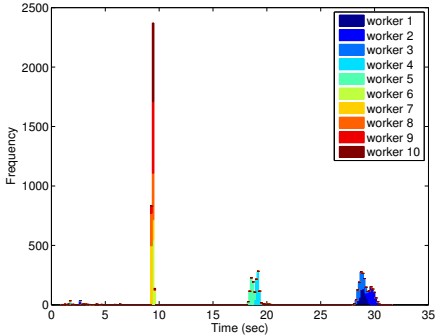

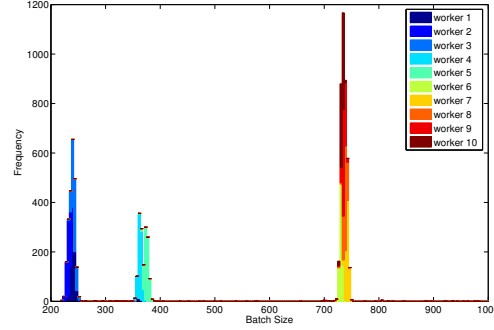

(a) FMB: number of batches completed by each worker versus time to complete each batch; batch size fixed.

(b) AMB: number of batches completed by each worker versus size of each batch; compute time fixed.

Figure 6: Histograms of worker performance in EC2 when stragglers are induced.

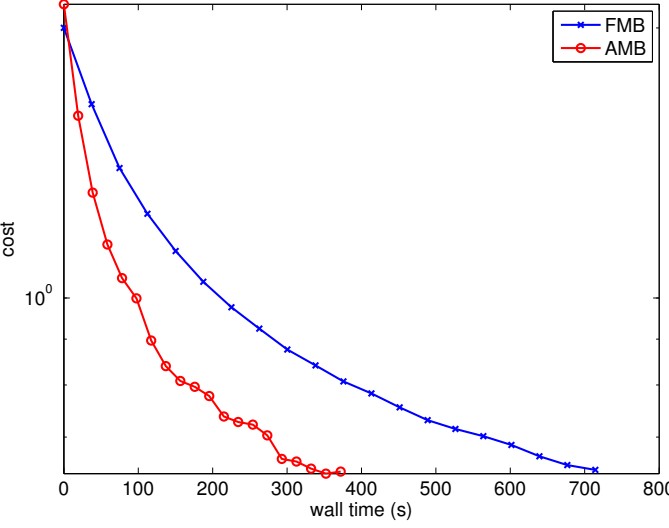

Figure 7: MNIST logistic regression performance of AMB and FMB on EC2 operating with induced straggler nodes.

is about 50%. Generally as the variation amongst stragglers increases we will see a corresponding improvement in AMB over FMB.

## I.4 PERFORMANCE WITH INDUCED STRAGGLERS ON AN HPC PLATFORM

We conducted another experiment on a high-performance computing (HPC) platform that consists of a large number of nodes. Jobs submitted to this system are scheduled and assigned to dedicated nodes. Since nodes are dedicated, no obvious stragglers exist. Furthermore, users of this platform do not know which tasks are assigned to which node. This means that we were not able to use the same approach for inducing stragglers on this platform as we used on EC2. In EC2, we ran background simulations on certain nodes to slow them down. But, since in this HPC environment we cannot tell where our jobs are placed, we are not able to place additional jobs on a subset of those same nodes to induce stragglers. Therefore, we used a different approach for inducing stragglers as we now explain.

First, we ran the MNIST classification problem using 51 nodes: one master and 50 worker nodes where workers nodes were divided into 5 groups. After each gradient calculation (in both AMB

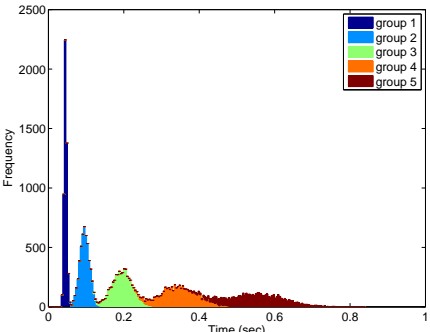 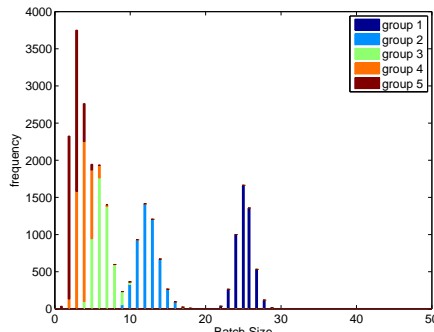

(a) FMB: number of batches completed by each worker versus time to complete each batch; batch size fixed.

(b) AMB: number of batches completed by each worker versus size of each batch; compute time fixed.

Figure 8: Histograms of worker performance in HPC when stragglers are induced.

and FMB), worker $i$ pauses its computation before proceeding to the next iteration. The duration of the pause of the worker in epoch $t$ after calculating the $s$-th gradient is denoted by $T_i(t,s)$. We modeled the $T_i(t,s)$ as independent of each other and each $T_i(t,s)$ is drawn according to the normal distribution $\mathcal{N}(\mu_j, \sigma_j^2)$ if worker $i$ is in group $j \in [5]$. If $T_i(t,s) < 0$, then there is no pause and the worker starts calculating the next gradient immediately. Groups with larger $\mu_j$ model worse stragglers and larger $\sigma_j^2$ models more variance in that straggler's delay. In AMB, if the remaining time to compute gradients is less than the sampled $T_i(t,s)$, then the duration of the pause is the remaining time. In other words, the node will not calculate any further gradients in that epoch but will pause till the end of the compute phase before proceeding to consensus rounds. In our experiment, we chose $(\mu_1, \mu_2, \mu_3, \mu_4, \mu_5) = (5, 10, 20, 35, 55)$ and $\sigma_j^2 = j^2$. In the FMB experiment, each worker calculated 10 gradients leading to a fixed minibatch size $b = 500$ while in AMB each worker was given a fixed compute time, $T = 115$ msec. which resulted in an empirical average minibatch size $b \approx 504$ across all epochs.

Figures 8a and 8b respectively depict the histogram of the compute time (including the pauses) for FMB and the histogram of minibatch sizes for AMB obtained in our experiment. In each histogram, five distinct distributions can be discerned, each representing one of the five groups. Notice that the fastest group of nodes has the smallest average compute time (the leftmost spike in Figure 8a) and the largest average minibatch size (the rightmost distribution in Figure 8b).

In Figure 9, we compare the logistic regression performance of AMB with that of FMB for the MNIST data set. Note that AMB achieves its lowest cost in 2.45 sec while FMB achieves the same cost only at 12.7 sec. In other words, the convergence rate of AMB is more than five times faster than that of FMB.

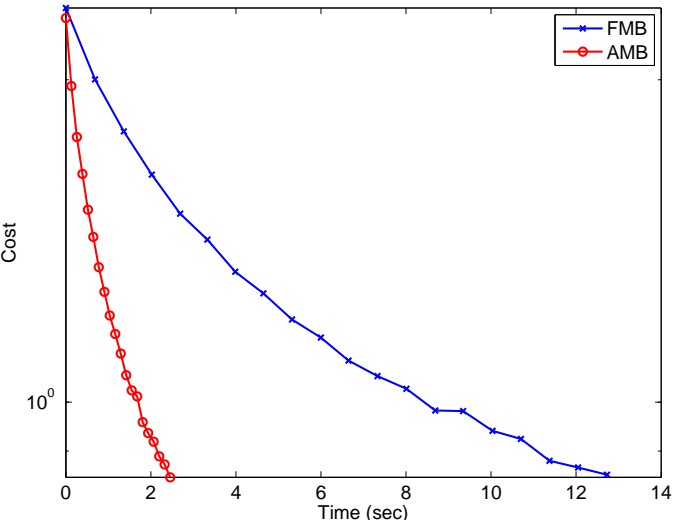

Figure 9: MNIST logistic regression performance of AMB and FMB on HPC operating with induced straggler nodes.

