# OpenReview forum: "ANYTIME MINIBATCH: EXPLOITING STRAGGLERS IN ONLINE DISTRIBUTED OPTIMIZATION"
_ICLR.cc/2019/Conference_

### Official Review · AnonReviewer2 · 2018-11-02
**Good paper with a nice idea**

**Rating:** 7
**Confidence:** 4

**Review:**

Summary:
The paper considers the problem of online stochastic convex optimization in a fully distributed topology. In particular, the authors focus on the synchronous setting and to avoid the slow progress that can be obtained by slow nodes, called stragglers, they propose an online distributed optimization method called Anytime Minibatch (AMB). In the update of AMB rather than fixing the minibatch size, they fix the computation time in each epoch. This characteristic prevents the stragglers from holding up the entire network, while allowing nodes to benefit from the partial work carried out by the slower nodes.

A convergence analysis of AMB is provided showing that the online regret achieves the optimum performance. Numerical evaluations where a comparison of AMB and the "Fixed MiniBatch" method (FMB)are also presented.

Comments:
I believe that the idea of the paper is interesting and the convergence analysis seems correct, however i have some concerns regarding  the presentation and the numerical evaluation.

1) In the title the word "online" is mentioned but never explained  in the main text. What is this mean? What are the differences compare to the "static" setting? See for example the work of [Tsianos, Rabbat (2016)] for more details on that. What are the related literature on this setting?

2) In the last paragraph of Introduction is highlighted that the algorithm AMB has the optimum performance?  The authors should add an appropriate reference there and explain why this is optimum for their setting. I believe that for the convenience of the reader current Section 5 called "previous work" can move immediately after introduction and more details of AMB with the existing literature should be provided. Probably rename the section "Closely relate work".

3) Section 2 is devoted mostly on the formal presentation of algorithm AMB. I strongly suggest the addition of a pseudocode of the algorithm in the appendix (or even in the main text if there is a space) where the reader can easily understand how the algorithm works.

4) On the Algorithm:  if some nodes are very slow and they do not make any update during the given time T what will happen? How this will affect the performance of the method? In this case does it make sense to increase the value of T.

5) On numerical evaluation:  A comparison of AMB and FMB  is presented both in synthetic and real data showing that AMB can be faster than FMB in terms of wall clock time.
I am not sure if the performance of the AMB is as good as one should expect especially for the case of synthetic data. Will it be possible to construct a synthetic example with extremely slow nodes where the improvement of the performance is much better than 50%?

In general i find the paper interesting, with nice ideas and I believe that will be appreciated from researchers that are interested on control theory/signal processing and information theory.  Since the paper is focused on convex optimization I am not sure if it will be particularly interesting for a substantial fraction of the ICLR attendees.

---

> ### Author Response · Authors · 2018-11-18
> **We thank the reviewer for the comments. In the following we address them.**
>
> 1) In our paper, “online” refers to the manner in which the nodes receive data samples: they obtain them one-by-one (or batch-by-batch).  Nodes do not have access to the entire dataset a priori. The online distributed learning problem has been studied extensively, and a few representative works (in addition to [Tsianos, Rabbat (2016)] and those already cited in our submission) are [Duchi et al. (2011)], [Dekel et al. (2012)], [Di Lorenzo, Scutari (2016)], and [Smith et al. (2018)]. Finally, we emphasize that AMB is not restricted to the online scenario.  We added the definition of online in the footnote of the first page. The algorithm can immediately be applied to settings where nodes are allocated predetermined data sets.
>
> 2) The optimum regret bound of O(\sqrt{m}) is well established in the stochastic approximation literature (see, e.g., [Nemirovski, Yudin (1983)] as well as [(Dekel et al. (2011)].   We show that AMB can achieve this bound in our theoretical analysis under appropriate conditions. We have revised our paper to include a reference and to clarify this point.
>
> ++We thank the reviewer for the suggestion of moving previous work. We have revised the paper per the reviewer’s suggestion.
>
> 3) We thank the reviewer for their suggestion.  We have included pseudocode in the revised appendix I (cf. Algorithm 1). Due to the space limitations, we did not find it possible to include the pseudocode in the main text.
>
> 4) Setting T to be very large is not a good choice as it will result in a large average minibatch size. We set T so that nodes compute the desired minibatch size in expectation. As the reviewer points out, by setting T this way we may sometimes encounter very slow nodes that are able to perform only very few computations. However, AMB is robust to these slow nodes as it supports variable sized minibatches in distinct epochs. The design of anytime minibatch is to have faster nodes compensate for slower nodes to get the minibatch close to the target expected minibatch size.
>
> 5) In response to the reviewer’s request, we have created an example to match the reviewer’s proposed scenario and in which the performance of AMB is even better.  Please see Appendix H.3 and Figures 6(a), 6(b), and 7 for the details and results. We conducted our new experiment on EC2 in which we created extra jobs on some nodes to slow those nodes to mimic stragglers. As is described in Appendix H, our setup of the experiment was as follows.  We first divided the nodes into three groups. In the first group, we ran two “interfering” simulations.  In the second group, we ran one interfering simulation. In the third group, we ran no extra simulations.  The purpose of these interfering simulations was to share the resources of those nodes with other computational tasks so that they are slowed down when running AMB or FMB.  The interfering jobs were matrix multiplications that were repeated continuously throughout the experiment.  As can be seen from the histograms of the compute times taken by the nodes to calculate an FMB gradient and the variable local minibatches processed by the nodes when running AMB presented in Figures 6(a) and 6(b), three distinct clusters of node processing speed are clearly visible.  The compute speeds of nodes in the first group is more than three times slower than those of the nodes in group 3. Figure 7 compares the convergence speed of AMB to FMB in this setting.  As can be seen from the figure, AMB is now more than twice as fast as is FMB.
>
> ++We acknowledge the reviewer’s point: the paper does have broad applications beyond representation learning. However, we point out that large-scale stochastic optimization is central to modern techniques for representation learning, including (e.g.) matrix factorization and deep networks, and thus we expect the anytime approach to be of substantial use to the ICLR community. Furthermore, many recent ICLR papers study optimization for machine and representation learning (two notable examples are [Kingma and Ba, (2015)] and [Chen et al., (2016)]. Finally, although the theoretical analysis is given for convex problems, the general approach can be applied to non-convex problems, with a reasonable expectation of similar performance improvements.

---

> > ### Author Response · Authors · 2018-11-25
> > **This response is a follow-up to our earlier response pertaining to the reviewer’s request.**
> >
> > We ran a different set of experiments on an HPC platform that consists of many nodes. On this platform, users are allocated dedicated nodes to run their jobs resulting in an almost straggler-free compute environment. Furthermore, users do not know which tasks are assigned to which nodes meaning that we were not able to use the same approach for inducing stragglers on this platform as we did on EC2. Hence, we used a different approach as we explain briefly here. For more details, the reader is referred to Appendix H.4. In this set of experiments, we divided the nodes into 5 groups. During the compute phase and after each gradient calculation, workers paused their calculations for some random amount of time. We modeled the pauses as independent random variables but not necessarily identically distributed. We used five different normal distributions to model the pauses of the five groups. See Figures 8(a) and 8(b) for the histograms of FMB compute time and AMB minibatches, respectively. We can observe five distinct distributions in both figures each representing one of the five groups. Our numerical results for this experiment in Figure 9 show the convergence rate of AMB is more than five times faster than that of FMB.

---

### Official Review · AnonReviewer1 · 2018-11-02
**Useful approach to mitigate stragglers in distributed computing setups**

**Rating:** 7
**Confidence:** 4

**Review:**

This paper studies distributed optimization in the presence of straggling computing nodes. In a synchronous distributed optimization approach, the stragglers delay the entire computation as the synchronization operation cannot be performed till every computing nod has completed its task. This paper aims to mitigate the effect of stragglers by proposing Anytime MiniBatch (AMB) approach, where each computing node is allowed to process the different number of samples between two synchronization steps. In particular, each node is given $T$ unit time to process as many samples as it can. After that, the nodes are allowed to aggregate the information among themselves through a consensus mechanism for another $T_c$ unit time. In contrast with this, the usual Fixed MiniBatch (FMB) approach requires each node to process a fixed number of samples before invoking aggregating step. The presence of stragglers can significantly increase the time between two synchronization step and slow down the overall optimization process.

This paper combines their AMB approach with the dual averaging method. The paper presents sample-path wise regret bounds for convex optimization under additional standard assumptions (e.g., Lipschitz continuousness, smoothness). The paper then compares analytically and experimentally compare the speed-ups obtained by their AMB approach as compared to the FMB approach. The paper studies an interesting problem and proposes a simple and practical solution. The paper is well written and makes novel contributions with sound analysis. The experimental evaluation on the real system also corroborates the theoretical findings.

Comments/questions:

The reviewer did not find the justification of using $c_i(t)$ in the definition of regret (cf. (17)) very clear. Is it because we also want compare with a centralized setting which does not have the communication overhead? As far as evaluation between distributed schemes (e.g., AMB, FMB etc) is concerned, shouldn't one define the regret with respect to $b_i(t)$s itself?

Can the authors comment on the setup where the communication links are also unpredictable and may experience congestion? In this case, one would encounter variable communication overhead to achieve the consensus error up to $epsilon$.

In Sec. 4, could authors comment on the settings where $O(sqrt(n-1))$ speed up is achievable?

Minor typos: In eq. (127)  E[S] -> S_F, S_T -> S_A. In eq. (128)  S_T -> S_A.

---

> ### Author Response · Authors · 2018-11-18
> **We thank the reviewer for the comments. In the following we address them.**
>
> ++There are two ways to think about whether to include the c_i(t) in the definition of regret.  The first is when comparing AMB to FMB, the second is when comparing AMB to the performance attained by a single machine (i.e., a non-distributed context in which there are no communication delays.  In the first case, both AMB and FMB suffer the same communication delays and one can consider either only b_i(t) – i.e., ignoring the communication delays since both schemes suffer the same delays – or, as we do, can consider c_i(t). Either approach will lead to the same design insight.  The second case is quite different.  In the single-processor setting, we should consider c_i(t), rather than b_i(t), as the loss in potential computation during communication times that is suffered by a distributed scheme is not suffered by a single-processor approach. Thus, by deriving our results with respect to c_i(t), rather than b_i(t), one gains the flexibility to compare our scheme to either FMB or to single-processor systems.  We note that a similar approach was taken by other researchers in various papers including [Tsianos, Rabbat (2016)][ Dekel et al. (2012)].
>
> ++In AMB, there is a fixed communication time T_c. This means that each node, in general, performs several rounds of consensus until the communication period of length T_c concludes. A node that suffers from network congestion will not hold up the entire process. Our choice of fixing T_c does make the number of rounds of consensus that occur in each consensus epoch variable.  This naturally leads to variability in the amount of consensus error in each round. One can choose T_c to be large enough to meet a target average level of consensus.
>
> ++There are distributions that achieve this (maximal) bound on the speed up. Such a distribution is given in [Bertsimas et al. (2006)] (after equation 10). Of course, not all distributions achieve that speed up. A standard distributed used to model computation time is, as already mentioned, the shifted exponential distribution.  In our paper, we show that for the shifted exponential distribution, the acceleration is log(n), which is lower than O(sqrt(n-1)). We have revised the paper to clarify this point further.
>
> ++We thank the reviewer for pointing out the typo. We corrected the typo in our revision.

---

### Official Review · AnonReviewer3 · 2018-11-03
**The contribution is not clear and significant for accelerating distributed algorithms**

**Rating:** 4
**Confidence:** 4

**Review:**

Overall, this paper is well written with clearly presentation.
However, the contribution is not good enough to research the ICLR requirement.
Although the authors propose some method to balance the computation between distributed workers, which should be important for distributed optimization algorithm design, but not enough numerical experiments are proposed to prove the efficiency.
Even though some convergence analysis is given.
The main concern of this paper is to significantly increase the algorithm efficiency, but both theoretical and numerical results are lack of strength.

---

> ### Author Response · Authors · 2018-11-18
> **We thank the reviewer for the comments. In the following we address them.**
>
> We briefly summarize the impact of the work as we see it. We have developed a method for distributed optimization that accounts for the real-world non-idealities of cloud computing systems, including variability in compute node throughput and load, network congestion, and hardware failure. This approach allows straggler nodes to make full use of their computations, and in numerical experiments is twice as fast as fixed minibatching. We provide a rigorous theoretical analysis of our approach applied to convex problems, showing that it can achieve the optimum regret bound and that it offers a speedup factor of up to \sqrt{n-1} over fixed minibatching. These performance improvements substantially accelerate the learning of representations from large-scale datasets, and therefore we believe our contribution will be of benefit to the ICLR community.
>
> Although the main paper has only a few simulations, we have more numerical results in the appendix (cf., App. H, Figure 3-5). Moreover, in the revised appendix, we have included additional simulations beyond those in the original submission. For example, in response to Reviewers 1’s comments we performed additional simulations that are found in the appendix (cf., Figure 7). We performed these new experiments on EC2 inducing the wider variety of straggler types requested by Reviewer 2.  In this environment, AMB is twice as fast as FMB, taking about half the time it takes FMB to achieve the same error rate.
>
> We welcome and will be happy to address in detail any further critiques of this work during the rebuttal period.

---

> > ### Author Response · Authors · 2018-11-25
> > **More numerical simulations added in the new revision**
> >
> > In addition to our response above, we would like to note the reviewer that we have additional results related to simulations under induced-stragglers environment. The reviewer is referred to our new Appendix H.4. (cf., Figure 9) for more details. We conducted this experiment on a high-performance computing (HPC) platform and observed that our method is up to five times faster than FMB.

---

### Meta-Review · Area_Chair1 · 2018-12-12
**Good paper**

**Confidence:** 3
**Recommendation:** Accept (Poster)

**Metareview:**

The reviewers that provided extensive reviews agree that the paper is well-written and contains solid technical material. The paper however should be edited to address specific concerns regarding theoretical and empirical aspects of this work.